# PHYLOTEXTDIFF: TEXT-BASED DISCRETE DIFFUSION FOR GENERATIVE PHYLOGENETIC INFERENCE

## ABSTRACT

Phylogenetic inference aims to reconstruct the evolutionary relationships among species from DNA sequence data. Despite its long history and broad applications, accurately modeling phylogenetic tree distributions remains challenging due to the combinatorial explosion of possible topologies. In this work, we introduce PhyloTextDiff, the first textual-based and discrete diffusion model for phylogenetic inference. PhyloTextDiff is trained once using multiple DNA matrices allowing it to learn common patterns both in the DNA sequences and the textual tree representations. It operates non-autoregressively, enabling fast and scalable generation that is minimally impacted by the number of taxa. Leveraging the diffusion process, PhyloTextDiff is particularly well-suited for exploring the vast and multimodal landscape of phylogenetic tree spaces. Experiments on benchmark datasets demonstrate that PhyloTextDiff produces high-quality trees and enables efficient exploration of large phylogenetic spaces, opening the door to large-scale phylogenetic discovery.

## 1 INTRODUCTION

Phylogenetic inference is a fundamental task in computational biology, aiming to reconstruct the evolutionary relationships among species using data such as DNA sequences, protein structures, or morphological traits. Accurate phylogenetic inference underpins diverse applications, including epidemiology (Dudas et al., 2017; Attwood et al., 2022), antibiotic resistance studies (Aminov and Mackie, 2007; Ranjbar et al., 2020; Layne et al., 2020), conservation genetics (DeSalle and Amato, 2004), and risk assessment of invasive species (Hamelin et al., 2022; Dort et al., 2023). Despite its importance, phylogenetic inference is notoriously difficult. Tree topologies are discrete while branch lengths are continuous, and the number of possible trees grows super-exponentially: for $n$ taxa (or species), there exist $(2n - 5)!!$ [1] distinct unrooted bifurcating tree topologies (Billera et al., 2000). Even under simplifying assumptions such as a molecular clock, maximum likelihood tree reconstruction remains NP-hard (Chor and Tuller, 2005).

Traditional approaches such as Markov Chain Monte Carlo (MCMC) sampling rely on Felsenstein's pruning algorithm (Felsenstein, 1981) to compute tree likelihoods and converge in theory to the true posterior (Huelsenbeck and Ronquist, 2001; Larget and Simon, 1999; Altekar et al., 2004; Xie et al., 2011; Höhna et al., 2016; Drummond and Rambaut, 2007; Nguyen et al., 2015; Goloboff et al., 2008; Stamatakis et al., 2005). In practice, however, MCMC suffers from slow convergence, strong correlations between successive states, and rare transitions between modes of the posterior (Tjelmeland and Hegstad, 2001; Neal, 1996). Variational inference (VI) methods provide faster alternatives. Subsplit Bayesian Networks (SBNs) (Zhang and Matsen, 2018; Zhang, 2023; 2020; Zhang and Iv, 2019) capture distributions over tree topologies but are limited by pre-specified structures. Other VI methods build trees incrementally (Xie and Zhang, 2023; Zhou et al., 2024a; Koptagel et al., 2023; Mimori and Hamada, 2023), which scales poorly with the number of taxa and does not guarantee improved accuracy. Both MCMC and VI approaches generally require dataset-specific retraining, further limiting their scalability. Recently, generative models have emerged as promising alternatives for phylogenetic inference. PhyloGen (Duan et al., 2024) employs a VAE-based framework grounded in the neighbor-joining algorithm. One notable approach is PhyloVAE (Xie et al., 2025), which offers a method for unsupervised phylogenetic inference by learning a continuous latent

---

[1]The double factorial of a number $n$, denoted by $n!!$, is the product of all the positive integers up to $n$ that have the same parity (odd or even) as $n$

representation of tree topologies. While PhyloVAE excels at capturing global structure, our diffusion model, **PhyloTextDiff**, focuses on refining the phylogenetic posterior distribution, addressing the critical need for improved sampling across the combinatorial tree space.

In this work, we introduce PhyloTextDiff, the first discrete diffusion and text-based generative model for phylogenetic inference. By operating on the Newick textual representation of trees, PhyloTextDiff directly models the discrete topology of phylogenetic trees while enabling fast, non-autoregressive generation. Unlike prior methods, our approach avoids the linear growth in sampling time with the number of taxa and efficiently explores multimodal tree distributions.

Our key contributions are:

- **First textual-based model for phylogenetics.** PhyloTextDiff operates directly on Newick trees, treating them as sequences of tokens. This textual representation allows the model to leverage powerful sequence modeling techniques from NLP, capture both local and global tree structures, and avoid combinatorial constraints that hinder conventional tree-based methods.

- **Cross-dataset training and representation learning.** Training on multiple datasets simultaneously allows PhyloTextDiff to learn a shared Newick representation, to capture common structural patterns across datasets, and to generalize effectively to unseen tree distributions.

- **Model-agnostic posterior refinement via diffusion**. By learning from trees generated by any phylogenetic tree sampling method, PhyloTextDiff leverages a discrete diffusion process to refine existing posteriors, discover additional high-probability modes, and generate a richer, more diverse set of candidate trees.

- **Empirical validation.** Experiments on benchmark datasets demonstrate that PhyloTextDiff achieves high-quality reconstructions, competitive likelihoods, and efficient exploration of large phylogenetic spaces.

## 2 RELATED WORK

**MCMC-based approaches.** Classical methods such as MrBayes (Huelsenbeck and Ronquist, 2001; Ronquist and Huelsenbeck, 2003; Ronquist et al., 2012), RevBayes (Höhna et al., 2016), and PAUP* (Swofford, 2003) remain the gold standard for phylogenetic inference. While theoretically convergent, MCMC suffers from slow mixing, correlations between samples, and poor coverage of low-probability modes, making accurate posterior estimation computationally expensive.

**Variational inference and Generative Models.** VI offers faster alternatives by approximating the posterior distribution. Subsplit Bayesian Networks (SBNs) (Zhang and Matsen, 2018) learn distributions over tree topologies but require pre-specified structures and scale poorly with the number of taxa. Recent incremental approaches such as ArTree (Xie and Zhang, 2023) and PhyloGFN (Zhou et al., 2024a) build trees sequentially but face growing computational costs as the number of taxa increases. More recently, deep generative methods have been explored. PhyloGen (Duan et al., 2024) adapts VAEs with neighbor-joining priors, while PhyloVAE (Xie et al., 2025) learns continuous latent representations of trees. These models improve scalability and representation learning but still struggle to efficiently explore the full posterior distribution.

## 3 BACKGROUND AND PROBLEM DEFINITION

### 3.1 REPRESENTATION OF PHYLOGENETIC TREES

A phylogenetic tree is represented as $(\tau, \mathbf{b})$, where the topology $\tau = (\mathcal{V}, \mathcal{E})$ is a bifurcating graph with nodes $\mathcal{V}$ and edges $\mathcal{E}$, and the vector of branch lengths $\mathbf{b}$ consists of non-negative values measuring evolutionary change. Trees may be *rooted*, with a unique ancestor node, or *unrooted*, which capture relatedness without direction of ancestry. In this work, we focus on unrooted trees, though our method extends to rooted trees. Leaf nodes (degree 1) correspond to observed species (or taxa), while internal nodes (degree 3) represent unobserved ancestors.

Trees can be compactly encoded in the *Newick format*, which uses nested parentheses and commas to describe branching structure, and terminates with a semicolon. Leaves are labeled by species names,

and both rooted and unrooted trees can be represented (Fig. 1). For a tree with $n$ taxa, the Newick string contains $n$ labels, $n-1$ commas, and one semicolon, plus $(n-1)$ pairs of parentheses in the rooted case or $(n-2)$ pairs in the unrooted case.

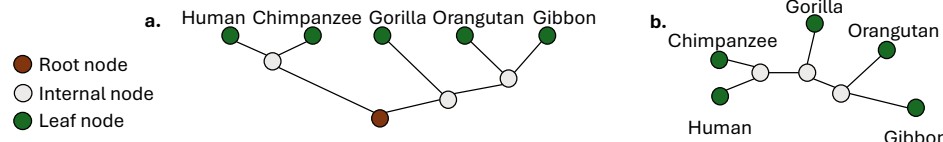

Figure 1: Examples of Newick representations for $n = 5$ taxa. (a) Rooted tree: `((Human,Chimpanzee),(Gorilla,(Orangutan,Gibbon)));` (b) Unrooted tree: `(Human,Chimpanzee,(Gorilla,(Orangutan,Gibbon)));`.

## 3.2 DISCRETE SCORE-BASED DIFFUSION MODELS

Diffusion models (Sohl-Dickstein et al., 2015) are latent-variable generative models defined by a forward process that gradually corrupts data $\boldsymbol{x}_0 \sim q_{\text{data}}$ into a base distribution $q_{\text{base}}$, and a reverse process $q_\theta$ that denoises back to the data distribution. For discrete state space $\mathcal{X} = \{1, \ldots, N\}^D$,

$$q(\boldsymbol{x}_{1:T} \mid \boldsymbol{x}_0) = \prod_{t=1}^{T} q(\boldsymbol{x}_t \mid \boldsymbol{x}_{t-1}), \quad q_\theta(\boldsymbol{x}_{0:T}) = q(\boldsymbol{x}_T) \prod_{t=1}^{T} q_\theta(\boldsymbol{x}_{t-1} \mid \boldsymbol{x}_t), \boldsymbol{x} \in \mathcal{X}. \quad (1)$$

Score-based diffusion learns the log-probability gradient $\nabla_{\boldsymbol{x}} \log p(\boldsymbol{x})$ with a score network $s_\theta : \mathbf{R}^D \to \mathbf{R}^D$ (Song and Ermon, 2019; Song et al., 2019). The forward process is defined by a continuous-time rate matrix $R_t$, such that $q_{t|t-\Delta t}(\mathbf{x}|\hat{\mathbf{x}}) = \delta_{\mathbf{x},\hat{\mathbf{x}}} + R_t(\hat{\mathbf{x}}, \mathbf{x})\Delta t + o(\Delta t)$(Austin et al., 2021).This process has a known reversal, given by the backward rate matrix $\bar{R}_t$ (Sun et al., 2023):

$$\frac{dq_{t-\Delta t}}{dt} = \bar{R}_t(\hat{\mathbf{x}}, \mathbf{x})\Delta t + o(\Delta t), \quad \bar{R}_t(\boldsymbol{x}, \hat{\boldsymbol{x}}) = \frac{q_t(\hat{\boldsymbol{x}})}{q_t(\boldsymbol{x})} R_t(\hat{\mathbf{x}}, \boldsymbol{x}). \quad (2)$$

Therefore, once we know the ratio $\left[\frac{q_t(\hat{\boldsymbol{x}})}{q_t(\boldsymbol{x})}\right]_{\hat{\boldsymbol{x}} \neq \boldsymbol{x}}$, we can obtain the generative flow towards $q_{\text{data}}$. Full technical details are relegated to Appendix B.

## 3.3 PROBLEM STATEMENT

We are provided with a set of DNA matrices $\boldsymbol{Y}_1, \boldsymbol{Y}_2, \cdots, \boldsymbol{Y}_N$, where $\boldsymbol{Y}_j \in \Sigma^{n_j \times m}$ and $\Sigma = \{A, C, G, T\}$. Each matrix represents an evolutionary set up with a set of taxa. We aim to learn to sample from the phylogenetic posterior distributions $p(\tau, \mathbf{b} \mid \boldsymbol{Y}_j), \forall j \in [1, N]$.

Although our work is applicable to any likelihood function that can be evaluated for each $(\tau, \boldsymbol{b})$, in our experimental work, we define the likelihood $p(\boldsymbol{Y}_j \mid \tau, \mathbf{b})$ (Eq 9) via a substitution model. Specifically, we adopt the Jukes-Cantor continuous-time substitution model (Jukes and Cantor, 1969)) that has commonly been employed in the phylogenetic inference literature (Zhou et al., 2024a; Zhang, 2023). More details about the likelihood computation can be found in Appendix A. As in most prior work, we assume independent priors $p(\tau, \boldsymbol{b}) = p(\tau)p(\boldsymbol{b})$, with a uniform prior on the tree topology $p(\tau)$ and exponential prior on the branch lengths $p(\boldsymbol{b})$. Evaluation of the normalization constant $p(\boldsymbol{Y}_j)$ is intractable, but the constant is the same for all trees within the dataset.

Our goals are: (i) efficiently sample from a distribution that matches the posterior as closely as possible; (ii) sample high-probability trees; and (iii) successfully explore multiple modes. Ideally, the approach should achieve high diversity, i.e., sampling a large number of unique trees. In order to measure the extent to which we achieve these goals, we evaluate lower bounds on the marginal log-likelihood (MLL) as well as several diversity measures, including Simpson's diversity metric.

**Generative Model** Our hypothesis is that common patterns exist across the provided DNA datasets and their textual tree representations. We are therefore motivated to learn a common generative model $q_{\theta,\phi}(\tau, \boldsymbol{b} \mid \boldsymbol{Y}_j)$ that can conditionally sample from the posterior $p(\tau, \boldsymbol{b}|\boldsymbol{Y}_j)$ for every $\boldsymbol{Y}_j$. We train the model using all of the $N$ available DNA matrices.

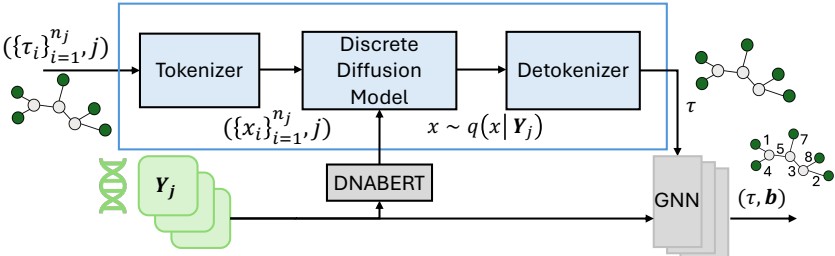

Figure 2: Framework of PhyloTextDiff. DNA matrices $Y_j$ and their associated set of trees in Newick format are fed to our discrete diffusion model. We use DNABERT to produce species-aware embeddings of the DNA. Our tree topology sampler is then trained to generate tokenized trees $x$ conditioned on $Y_j$, which are subsequently detokenized and passed to GNNs to infer branch lengths.

## 4 PROPOSED METHOD

In this section, we introduce PhyloTextDiff, a discrete diffusion model for phylogenetic inference. Our methodology builds on the ability of existing phylogenetic sampling methods to generate samples from an *approximate* posterior distribution for a single DNA matrix. This means that we can use, for example, MrBayes (Ronquist et al., 2012) to produce samples from $\tilde{p}_j(\tau, b \mid Y_j)$, where $\tilde{p}_j$ is an approximation to the true posterior. Given a set of samples $\{\tau_j^{(i)}\}_{i=1}^{n_j} \sim \tilde{p}_j$ for each $j = 1, \ldots, N$, we then train a common discrete diffusion model $q_\theta(\tau \mid Y_j)$ that learns not only from these samples, but also from DNABERT encoded representations of the DNA matrices. The model also operates on tokenizations of the textual representations of the trees and can thus identify commonalities in the textual structure. The model is trained to perform conditional sampling, but because the model parameters $\theta$ are common for all $Y_j$, the model can (and is encouraged to) capture common structure across the DNA matrices and textual representations.

### 4.1 PHYLOTEXTDIFF: AN OVERVIEW

The high-level structure of PhyloTextDiff is depicted in Figure 2. We introduce an approximation of the posterior $q_{\theta,\phi}(\tau, \mathbf{b} \mid Y_j) = q_\theta(\tau \mid Y_j) q_{\phi,j}(\mathbf{b} \mid \tau, Y_j)$. We use a diffusion model to learn to sample from the approximate distribution of the tree topologies $q_\theta(\tau \mid Y_j)$. This diffusion model is trained once for all the datasets. Conditioned on a sample $\tau^{(k)}$ generated by the discrete diffusion module, a GNN-based module is then employed to learn and sample from $q_{\phi,j}(\mathbf{b} \mid \tau, Y_j)$. The GNN-based module is trained once per dataset.

The first step in our process is to collect tree topology samples $\{\tau_j^{(i)}\}_{i=1}^{n_j} \sim \tilde{p}_j$ for each $j = 1, \ldots, N$ using a base phylogenetic sampler. In our experiments, we explore performance when using a range of different models. We then represent these tree topologies in a text-based Newick format and tokenize them using a custom Phylogenetic Tokenizer. These tokenized representations are fed to our Discrete Diffusion Model (Section 4.2). The Discrete Diffusion Model is provided with DNABERT-S (Zhou et al., 2024b) embeddings of the DNA matrices so that it can learn to sample form conditional distributions $q_\theta(\tau \mid Y_j)$ (which are variational approximations of the marginal posteriors).

The branch length distribution $q_{\phi,j}(\mathbf{b} \mid \tau, Y_j)$ is approximated using GNNs, following directly Zhang (2023). We model the branch length distribution as a diagonal lognormal distribution and parametrize $q_\phi(\mathbf{b}|\tau, Y)$ using the learnable topological features with the edge convolution operator (EDGE) for GNNs. This is not a novel contribution, but for completeness the approach is detailed in Appendix E.

### 4.2 DISCRETE DIFFUSION MODEL

**Tokenizer.** We collect tree topology samples $\{\tau_j^{(i)}\}_{i=1}^{n_j} \sim \tilde{p}_j$ in textual format for each dataset $j = 1, \ldots, N$ using a base phylogenetic sampler. We introduce a custom tokenizer that converts textual phylogenetic trees into fixed-length token sequences. For datasets $1, \ldots, N$ with $n_1, \ldots, n_N$ taxa, let $n_{\max} = \max_j n_j$ be the maximum number of taxa. Each tree with $n$ taxa is mapped to a sequence of length $d = 4n_{\max} - 4$, augmented with start, end, and padding tokens to standardize

sequence length across datasets. The vocabulary $\mathcal{W}_{1:N}$ contains all tokens: four special symbols, four Newick structural characters, and one token per taxon name per dataset. Unlike standard tokenizers, our approach assigns fixed indices to taxa names, eliminating the need for learned segmentation. More details can be found in Appendix C.

**Forward and Backward Process.** The tokenized trees are then fed to our diffusion model. We model our probability distributions over the support $\mathcal{X} = [1, \cdots, |\mathcal{W}_{1:N}|]$. Our method is built upon a continuous time process from $t = 0$ to $t = T$. We define a Continuous Time Markov Chain (CTMC) from $q_0 \approx q_{data}$ to $q_T \approx q_{base}$, the stationary distribution of the CTMC which contains only noise. Motivated by the success of BERT (Devlin et al., 2019) and masked language models, we consider a transition rate matrix with an absorbing state, such that each token either stays the same or transitions to the absorbing state with probability $\sigma_t$ that increases with $t$:

$$\boldsymbol{R}_t = \sigma_t \boldsymbol{R}, \quad \boldsymbol{R} = \left( \boldsymbol{e}^{(|\mathcal{W}_{1:N}|+1)} - I \right) \in \mathbb{R}^{(|\mathcal{W}_{1:N}|+1) \times (|\mathcal{W}_{1:N}|+1)}, \tag{3}$$

where $\boldsymbol{e}^{(|\mathcal{W}_{1:N}|+1)}$ is the one-hot encoding of the absorbing state. We condition the time reversal of the forward process on the DNA matrix $\boldsymbol{Y}_j \in \Sigma^{n_j \times m}$. The backward process is also a CTMC with initial distribution $q_T(\boldsymbol{x}_T \mid \boldsymbol{Y}_j)$ at $t = T$ and target distribution $q_0(\boldsymbol{x}_0 \mid \boldsymbol{Y}_j)$ and is entirely determined by its forward and backward rate matrices $\boldsymbol{R}$ and $\bar{\boldsymbol{R}}$.

$$\bar{R}_t(\boldsymbol{x}, \hat{\boldsymbol{x}} \mid \boldsymbol{Y}_j) = \frac{q_t(\hat{\boldsymbol{x}} \mid \boldsymbol{Y}_j)}{q_t(\boldsymbol{x} \mid \boldsymbol{Y}_j)} R_t(\hat{\boldsymbol{x}}, \boldsymbol{x}), \quad \bar{R}_t(\boldsymbol{x}, \boldsymbol{x} \mid \boldsymbol{Y}_j) = - \sum_{\mathbf{z} \neq \boldsymbol{x}} \bar{R}_t(\mathbf{z}, \boldsymbol{x} \mid \boldsymbol{Y}_j), \quad \forall \hat{\boldsymbol{x}} \neq \boldsymbol{x}. \tag{4}$$

**Factorization of diffused state conditioned on $\boldsymbol{x}_0$.** We consider sequences $\boldsymbol{x} = (x^{(1)} \dots x^{(d)}) \in \mathcal{X}$. Since this is a continuous time process and each dimension's forward process is independent of the others, we can factorize the forward process such that each dimension propagates independently. The joint distribution is entirely determined by its conditional probabilities, so we follow previous works (Lou et al., 2024; Campbell et al., 2022) and only evaluate ratios with Hamming distance of one. Each token in the sequence $\boldsymbol{x}$ is perturbed independently with a noise level $\sigma_t$, i.e.,

$$q_{t|0}^{seq}(\hat{\boldsymbol{x}}|\boldsymbol{x}) = \prod_{i=1}^{d} q_{t|0}^{tok}(\hat{x}^{(i)}|x^{(i)}), \quad \text{s.t.} \quad q_{t|0}^{tok}(\hat{x}^{(i)}|x^{(i)}) = \exp\left( \int_0^t \sigma(s)ds R(\cdot, \cdot) \right). \tag{5}$$

**Score learning.** In order to derive the backward rate matrix $\bar{\boldsymbol{R}}$, we propose to learn the conditioned concrete score $s_\theta(\boldsymbol{x}, t \mid \boldsymbol{Y}_j) \approx \left[ \frac{q_t(\mathbf{z}|\boldsymbol{Y}_j)}{q_t(\boldsymbol{x}|\boldsymbol{Y}_j)} \right]_{\mathbf{z} \neq \boldsymbol{x}}$ $(s_\theta : \mathcal{X} \times \mathbb{R} \times \Sigma^{n_j \times m} \rightarrow \mathbb{R}^{\mathcal{X}})$ using a modified version of the diffusion weighted denoising score entropy loss (Lou et al., 2024) conditioned on the DNA $\boldsymbol{Y}_j$. With $K(a) = a(\log a - 1)$, it is defined as:

$$\mathcal{L}_\theta(\boldsymbol{x}_0 \mid \boldsymbol{Y}_j) = \int_0^T \mathbb{E}_{\boldsymbol{x}_t \sim q_{t|0}(\cdot|\boldsymbol{x}_0)} \sum_{\mathbf{z} \neq \boldsymbol{x}_t} R_t(\boldsymbol{x}_t, \mathbf{z}) l_t(\boldsymbol{x}_t, \mathbf{z} \mid \boldsymbol{x}_0, \boldsymbol{Y}_j) dt, \tag{6}$$

s.t. $l_t(\boldsymbol{x}_t, \mathbf{z} \mid \boldsymbol{x}_0, \boldsymbol{Y}_j) = \left( s_\theta(\boldsymbol{x}_t, t \mid \boldsymbol{Y}_j)_{\mathbf{z}} - \frac{q_{t|0}(\mathbf{z}|\boldsymbol{x}_0)}{q_{t|0}(\boldsymbol{x}_t|\boldsymbol{x}_0)} \log s_\theta(\boldsymbol{x}_t, t \mid \boldsymbol{Y}_j)_{\mathbf{z}} + K\left( \frac{q_{t|0}(\mathbf{z}|\boldsymbol{x}_0)}{q_{t|0}(\boldsymbol{x}_t|\boldsymbol{x}_0)} \right) \right).$

**Sampling.** Discrete Tweedie's Theorem (Lou et al., 2024) expresses the true denoiser in terms of ratios of forward probabilities. Since in practice only ratios between Hamming distance–1 sequences are accessible, we follow Lou et al. (2024) and employ, with $\sigma_t^{\Delta t} = \int_{t-\Delta t}^t \sigma(s)ds$, the $\tau$-leaping Tweedie denoiser:

$$q_{t-\Delta t|t}^{\text{tweedie}}(\boldsymbol{x}_{t-\Delta t}^{(i)} \mid \boldsymbol{x}_t^{(i)}, \boldsymbol{Y}_j) = \left( \exp(-\sigma_t^{\Delta t}\boldsymbol{R}), s_\theta(\boldsymbol{x}_t, t, \boldsymbol{Y}_j)_i \right)_{\boldsymbol{x}_{t-\Delta t}^{(i)}} \exp(\sigma_t^{\Delta t}\boldsymbol{R})(\boldsymbol{x}_t^{(i)}, \boldsymbol{x}_{t-\Delta t}^{(i)}). \tag{7}$$

**Adaptation to a structured space.** Phylogenetic tree topologies are subject to strict syntactic rules: for $n_j$ taxa, each valid Newick string contains $n_j - 2$ opening and closing parentheses, $n_j - 1$ commas, and a final semicolon, for a total length $d_j = 4(n_j - 1)$. To ensure that our diffusion model respects these constraints, we restrict the sampling space by fixing the start, end and padding tokens, while keeping intermediate positions masked before denoising. This significantly improves sample quality.

Figure 3: Overview of the PhyloTextDiff architecture. The input $x_t \sim q_{t|0}(x_0)$ is first embedded and enriched with DNA embeddings $\mathbf{M}$, then combined with rotary positional encodings and sinusoidal timestep embeddings $\sigma_t$. These representations are processed through DDiT blocks with cross-attention and the DDiT final layer to produce the predicted score $s_\theta(\boldsymbol{x}, t \mid \boldsymbol{Y}_j)$.

**Architecture.** In order to condition the backward diffusion process on the DNA sequences, we gather pre-trained embeddings of the DNA sequences using DNABERT-S (Zhou et al., 2024b), a genome foundation model that produces species-aware embeddings. For $N$ datasets with DNA matrices $\boldsymbol{Y}_1, \ldots, \boldsymbol{Y}_N$, we build an embedding matrix $\mathbf{M} \in \mathbb{R}^{(|\mathcal{W}_{1:N}|+1) \times h}$, where each row $i$ corresponds to the DNABERT-S embedding of a site if $i$ is a taxa token or to the zero vector $\mathbf{0}_h$ otherwise. We then condition our model by introducing cross-attention layers to enable dataset-aware guided diffusion. Our architecture is based on the Discrete Diffusion Transformer (DDiT) (Peebles and Xie, 2023) and a schematic overview is shown in Fig. 3. The full architecture is described in Appendix F.

To enrich the token representations, we integrate DNA information directly into the embeddings. Specifically, each token $x_t^{(i)}$ is embedded as the sum of its learnable embedding $\mathbf{E}_{x_t^{(i)}}$ and the corresponding row of $\mathbf{M}$, $\mathbf{M}_{x_t^{(i)}}$, i.e., $\mathbf{h}_t^{(i)} = \mathbf{E}_{x_t^{(i)}} + \mathbf{M}_{x_t^{(i)}}$. This approach allows the model to leverage both ground-truth biological information about the taxa and learnable features that capture the structure of the phylogenetic tree during the diffusion process.

### 4.3 MARGINAL LOG-LIKELIHOOD ESTIMATION

To assess how well PhyloTextDiff approximates the true posterior distribution, we derive the following importance-weighted variational lower bound on the marginal log-likelihood:

$$\mathbb{E}_{\substack{\boldsymbol{x}_{0:T} \sim q_{R,j} \\ \boldsymbol{b} \sim q_\phi(\boldsymbol{b}|\boldsymbol{x}_0)}} \log \sum_{\substack{i=1 \\ \boldsymbol{x}_{0:T} \sim q_{R,j} \\ \boldsymbol{b} \sim q_\phi(\boldsymbol{b}|\boldsymbol{x}_0)}}^{K} \frac{1}{K} P(\boldsymbol{Y}|\boldsymbol{x}_0, \boldsymbol{b}) p(\boldsymbol{x}_0) \frac{q_F(\boldsymbol{x}_{1:T}|\boldsymbol{x}_0)}{q_{R,j}(\boldsymbol{x}_{0:T})} \frac{p(\boldsymbol{b}|\boldsymbol{x}_0)}{q_\phi(\boldsymbol{b}|\boldsymbol{x}_0)} \leq \log P(\boldsymbol{Y}). \quad (8)$$

where $q_F(\boldsymbol{x}_{0:T}) = p_{\text{data}}(\boldsymbol{x}_0) \prod_{t=1}^{T} q_{t|t-1}(\boldsymbol{x}_t|\boldsymbol{x}_{t-1})$ denotes the forward process, $q_{R,j}(\boldsymbol{x}_{0:T}) = p_{\text{base}}(\boldsymbol{x}_T) \prod_{t=1}^{T} q_{t-1|t}(\boldsymbol{x}_{t-1}|\boldsymbol{x}_t, j)$ the backward process, $\boldsymbol{x}_{0:T} \sim q_{R,j}$ and $\boldsymbol{b} \sim q_{\phi,j}(\boldsymbol{b} \mid \boldsymbol{x}_0)$ denote independent samples from the backward diffusion process and the branch length model, respectively. This bound provides a principled way to approximate $\log p(\boldsymbol{Y}_j)$ while leveraging multiple samples for tighter estimation. The derivation of this bound is provided in Appendix D.

## 5 EXPERIMENTS

We evaluate PhyloTextDiff on eight real world benchmark datasets that are standard in the literature, that we call DS1-DS8 (Hedges et al., 1990; Garey et al., 1996; Yang and Yoder, 2003; Henk et al., 2003; Lakner et al., 2008; Zhang and Blackwell, 2001; Yoder and Yang, 2004; Rossman et al., 2001). These datasets consist of sequences from 27 to 64 eukaryote species with 378 to 2520 site observations. Details about the datasets can be found in Appendix G, and details about training in Appendix H. In the following sections, we present our results on Bayesian phylogenetic inference; we analyze the tree topological diversity as well as the coverage of posterior modes.

**Experimental Setup.** PhyloTextDiff is trained from trees that are sampled using another phylogenetic tree sampling approach. We conduct experiments using trees obtained using (i) the MCMC-based

Table 1: MLL estimates mean ($\uparrow$) and (variance ($\downarrow$)) on eight benchmark datasets. Best values are highlighted in green, second highest in blue, and third highest from in brown.

| Methods | Dataset | DS1 | DS2 | DS3 | DS4 | DS5 | DS6 | DS7 | DS8 |
|---|---|---|---|---|---|---|---|---|---|
| | #Taxa (N) | 27 | 29 | 36 | 41 | 50 | 50 | 59 | 64 |
| MCMC-based | MrBayes | -7108.42 (0.18) | -26367.57 (0.48) | -33735.44 (0.50) | -13330.44 (0.54) | -8214.51 (0.28) | -6724.07 (0.86) | -37332.76 (2.42) | -8649.88 (1.75) |
| Structure Representation | SBN | -7108.41 (0.15) | -26367.71 (0.08) | -33735.09 (0.09) | -13329.94 (0.20) | -8214.62 (0.40) | -6724.37 (0.43) | -37331.97 (0.28) | -8650.64 (0.50) |
| | VBPI-GNN (EDGE) | -7108.41 (0.14) | -26367.73 (0.07) | -33735.12 (0.09) | -13329.94 (0.09) | -8214.64 (0.38) | -6724.37 (0.40) | -37332.04 (0.12) | -8650.65 (0.45) |
| Structure Generation | phi-CSMC | -7290.36 (7.23) | -30568.49 (31.34) | -33798.06 (6.62) | -13582.24 (35.08) | -8367.51 (8.87) | -7013.83 (16.99) | NA | -9209.18 (18.03) |
| | GeoPhy | -7111.55 (0.07) | -26379.48 (11.60) | -33757.79 (8.07) | -13342.71 (1.61) | -8240.87 (9.80) | -6735.14 (2.64) | -37377.86 (29.48) | -8663.51 (6.85) |
| | GeoPhy LOO(3)+ | -7116.09 (10.67) | -26368.54 (0.12) | -33735.85 (0.12) | -13337.42 (1.32) | -8233.89 (6.63) | -6735.9 (1.13) | -37358.96 (13.06) | -8660.48 (0.78) |
| | ARTree | -7108.41 (0.19) | -26367.71 (0.09) | -33735.09 (0.09) | -13329.94 (0.17) | -8214.59 (0.34) | -6724.37 (0.46) | -37331.95 (0.27) | -8650.61 (0.48) |
| | PhyloGFN | -7108.95 (0.06) | -26368.9 (0.28) | -33735.6 (0.35) | -13331.83 (0.19) | -8215.15 (0.20) | -6730.68 (0.54) | -37359.9 (1.14) | -8654.76 (0.19) |
| | PhyloGen | -6910.02 (0.07) | -26257.09 (0.06) | -33481.57 (0.10) | -13063.15 (1.34) | -7928.4 (0.23) | -6330.21 (0.31) | -36838.42 (12.03) | -8171.04 (0.96) |
| Textual Representation | MrBayes + PhyloTextDiff | -6424.14 (3.62) | -25751.56 (2.62) | -33028.06 (6.32) | -12594.27 (4.71) | -7451.55 (2.36) | -5967.69 (3.99) | -36577.11 (4.84) | -7852.22 (5.36) |
| | IQ-Tree + PhyloTextDiff | -6303.89 (4.41) | -25571.31 (4.12) | -32934.96 (5.37) | -12497.71 (5.37) | -7418.71 (5.38) | -5930.22 (6.76) | -36540.99 (3.93) | -7851.44 (4.72) |
| | VBPI-GNN + PhyloTextDiff | -6298.56 (3.32) | -25560.36 (3.55) | -32923.35 (5.62) | -12514.25 (7.69) | -7409.00 (4.21) | -5926.91 (2.93) | -36533.82 (3.89) | -7840.57 (4.51) |

method MrBayes (Ronquist et al., 2012); (ii) the variational inference approach VBPI-GNN (Zhang, 2023); and (iii) IQ-TREE (Minh et al., 2020) with the ultrafast bootstrap UFBoot procedure (Minh et al., 2013; Hoang et al., 2018). Details of the sampling procedures are provided in Appendix M. To train the branch length model, we optimize a simplified version of the multi-sample lower bound described in Eq. 8 with $K = 10$, where we omit the computation of $q_R, j$ and $q_F$. During evaluation, we do not apply the space restriction trick described in Section 4.2, as it alters the final distribution. We use $T = 1024$ denoising steps for sampling during the edge model's training and $T = 2048$ steps for evaluation (see Appendix K for a sensitivity study on the denoising step). Our adopted model is the same as in (Zhou et al., 2024a; Zhang, 2023; Mimori and Hamada, 2023): (i) decomposed prior $p(\tau, \boldsymbol{b}) = p(\tau)p(\boldsymbol{b})$; (ii) uniform prior on the tree topology; (iii) exponential prior ($\lambda = 10$) on the branch lengths; (iv) Jukes-Cantor substitution model (Jukes and Cantor, 1969).

**Baselines and Performance Metrics.** We compare our method to the MCMC-based method MrBayes (Ronquist et al., 2012), the structure representation methods SBN (Zhang and Iv, 2019) and VBPI-GNN (Zhang, 2023), and the structure generation methods VaiPhy (Koptagel et al., 2023), GeoPhy and GeoPhy LOO(3)+ (Mimori and Hamada, 2023) , Artree (Xie and Zhang, 2023), PhyloGFN (Zhou et al., 2024a) and PhyloGen (Duan et al., 2024). We report the evidence lower bound (ELBO) ($K = 1$, $n_{\text{runs}} = 10$, $n_{\text{repetitions}} = 10$) and the marginal log-likelihood (MLL) estimate ($K = 1000$, $n_{\text{runs}} = 1$, $n_{\text{repetitions}} = 10$). For our method, we use the lower bound in equation 8. Details about the MLL lower bounds for the other methods are provided in Appendix I.

## 5.1 RESULTS AND DISCUSSION

**MLL bounds and ELBO.** Table 1 reports the MLL lower bounds and Table 2 reports the ELBO values. Across all datasets, PhyloTextDiff achieves the highest lower bound, with substantial improvements over both MCMC-based and variational baselines. On average, our estimates are higher by several hundred nats compared to the best-performing baselines. Traditional MCMC and subsplit methods generally trail behind, while autoregressive and GFlowNet and VAE inspired approaches (e.g., PhyloGen, PhyloGFN) show partial gains but remain consistently outperformed.

**Tree Topological Diversity Analysis.** We evaluate the diversity of sampled tree topologies using three complementary metrics. Details concerning the experimental procedure are provided in Appendix J.1. *Topological diversity* is quantified using Simpson's diversity index (He and Hu, 2005),

defined as $1-D$ (higher is better), with $D = \sum_i \frac{n_i(n_i-1)}{N(N-1)} \in [0,1]$, where $n_i$ denotes the number of trees with topology $i$ and $N$ is the total number of trees. $D$ measures the probability that two randomly sampled trees share the same topology. We also report two additional metrics: the *top frequency*, i.e., the proportion of the most frequently sampled topology (lower is better), and the *top 95% frequency*, defined as the number of distinct topologies accounting for 95% of the samples (higher is better), which reflects broader posterior support.

Table 2: Comparison of ELBO ($\uparrow$) and (variance ($\downarrow$)) on eight benchmark datasets. Best values are highlighted in green, second highest in blue, and third highest from in brown.

| Methods | Dataset | DS1 | DS2 | DS3 | DS4 | DS5 | DS6 | DS7 | DS8 |
|---|---|---|---|---|---|---|---|---|---|
| | #Taxa (N) | 27 | 29 | 36 | 41 | 50 | 50 | 59 | 64 |
| Structure Representation | SBN | -7110.24 (0.03) | -26368.88 (0.03) | -33736.22 (0.02) | -13331.83 (0.02) | -8217.80 (0.04) | -6728.65 (0.04) | -37334.85 (0.03) | -8655.05 (0.04) |
| | VBPI-GNN (EDGE) | -7110.26 (0.10) | -26368.84 (0.09) | -33736.25 (0.08) | -13331.80 (0.10) | -8217.80 (0.12) | -6728.57 (0.16) | -37334.84 (0.14) | -8655.01 (0.14) |
| Structure Generation | ARTree | -7110.09 (0.04) | -26368.78 (0.07) | -33735.25 (0.08) | -13330.27 (0.05) | -8215.34 (0.04) | -6725.33 (0.06) | -37332.54 (0.13) | -8651.73 (0.05) |
| | PhyloGen | -7005.98 (0.06) | -26362.75 (0.12) | -33430.94 (0.34) | -13113.03 (3.67) | -8053.23 (2.58) | -6324.9 (1.26) | -36838.42 (1.97) | -8409.06 (1.07) |
| **Textual Representation** | **MrBayes + PhyloTextDiff** | -6457.12 (0.88) | -25798.33 (2.63) | -33073.93 (1.13) | -12638.93 (1.71) | -7489.49 (1.67) | -6010.54 (1.44) | -36696.14 (8.89) | -7894.35 (1.59) |
| | **IQ-Tree + PhyloTextDiff** | -6356.79 (1.81) | -25667.39 (4.85) | -33012.91 (3.99) | -12576.35 (2.06) | -7473.54 (1.65) | -6007.33 (5.92) | -36648.47 (5.36) | -7906.63 (1.20) |
| | **VBPI-GNN + PhyloTextDiff** | -6341.83 (1.42) | -25632.69 (5.68) | -32976.97 (1.71) | -12570.64 (2.50) | -7455.93 (1.48) | -5973.61 (1.07) | -36620.67 (4.34) | -7887.46 (2.33) |

Table 3: Tree Topological Diversity for all datasets (DS1–DS8). Higher Diversity Index and Top 95% Frequency indicate more diverse posterior distributions.

| Metrics | Methods | DS1 | DS2 | DS3 | DS4 | DS5 | DS6 | DS7 | DS8 |
|---|---|---|---|---|---|---|---|---|---|
| **Simpson Diversity Index ($\uparrow$)** | MrBayes | 0.87268 | 0.68390 | 0.68227 | 0.89930 | 0.99900 | 0.99937 | 0.98802 | 0.97974 |
| | MrBayes + PhyloTextDiff | **0.99963** | 0.92847 | 0.98934 | **0.99954** | 0.99960 | 0.99965 | **0.99972** | **0.99963** |
| | IQ-TREE | 0.99743 | 0.98406 | 0.99722 | 0.99625 | 0.99851 | 0.99713 | 0.99714 | 0.99873 |
| | IQ-TREE + PhyloTextDiff | 0.99899 | **0.99866** | **0.99913** | 0.99909 | 0.99950 | 0.99971 | 0.99968 | 0.99934 |
| | VBPI-GNN | 0.96245 | 0.70976 | 0.81314 | 0.94849 | **0.99989** | **0.99998** | 0.99119 | 0.99464 |
| | VBPI-GNN + PhyloTextDiff | 0.99810 | 0.97908 | 0.99369 | 0.99843 | 0.99927 | 0.99965 | 0.99859 | 0.99806 |
| **Top Frequency ($\downarrow$)** | MrBayes | 0.279 | 0.505 | 0.454 | 0.268 | 0.007 | 0.005 | 0.035 | 0.116 |
| | MrBayes + PhyloTextDiff | **0.006** | 0.095 | 0.022 | **0.006** | 0.005 | 0.004 | **0.004** | **0.005** |
| | IQ-TREE | 0.012 | 0.051 | 0.014 | 0.021 | 0.014 | 0.019 | 0.018 | 0.008 |
| | IQ-TREE + PhyloTextDiff | 0.009 | **0.008** | **0.007** | 0.010 | 0.009 | 0.005 | **0.004** | 0.010 |
| | VBPI-GNN | 0.091 | 0.463 | 0.309 | 0.186 | **0.004** | **0.003** | 0.029 | 0.054 |
| | VBPI-GNN + PhyloTextDiff | 0.010 | 0.050 | 0.021 | 0.010 | 0.006 | 0.006 | 0.010 | 0.012 |
| **Top 95% Frequency ($\uparrow$)** | MrBayes | 41 | 5 | 12 | 111 | 629 | 698 | 240 | 289 |
| | MrBayes + PhyloTextDiff | **796** | 14 | 100 | **782** | 797 | 796 | **830** | **815** |
| | IQ-TREE | 450 | 243 | 424 | 402 | 614 | 513 | 475 | 593 |
| | IQ-TREE + PhyloTextDiff | 666 | **596** | **682** | 673 | 470 | 671 | 814 | 514 |
| | VBPI-GNN | 138 | 5 | 22 | 234 | **907** | **942** | 288 | 613 |
| | VBPI-GNN + PhyloTextDiff | 510 | 61 | 221 | 550 | 716 | 818 | 579 | 566 |

Table 3 compares PhyloTextDiff to each method it was trained on. Except for DS5 and DS6, PhyloTextDiff consistently enhances the topological diversity, for all the metrics, demonstrating a broader exploration of the posterior distribution.

**Unnormalized posterior comparison with MrBayes.** We compare the posterior distributions of MrBayes + PhyloTextDiff and MrBayes on dataset DS1. Details concerning the experimental procedure are provided in Appendix J.2. We observe that PhyloTextDiff produces a more dispersed posterior distribution compared to MrBayes, indicating greater sampling diversity (Fig. 4). Its posterior is also concentrated in higher-probability regions, suggesting more accurate identification of likely trees, as showcase by the results illustrated in the tables.

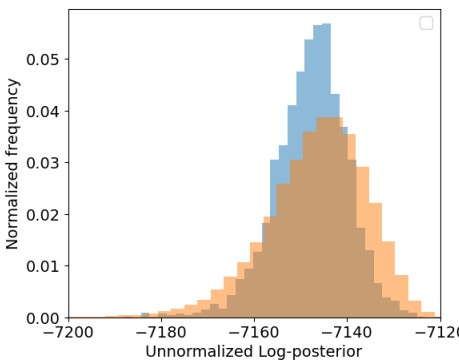

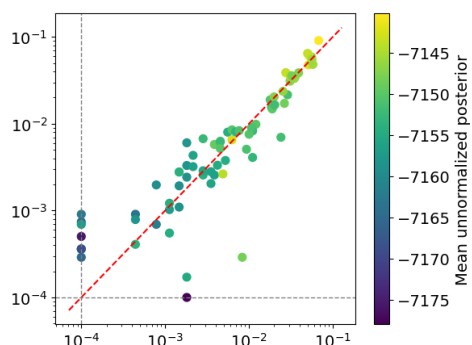

Figure 4: Normalized histograms of unnormalized log-posterior values for MrBayes (blue) and PhyloTextDiff (orange) on dataset DS1. PhyloTextDiff (21h) shows a less peaked distribution, indicating greater sampling diversity and better coverage of high-probability trees.

Figure 5: Tree space comparison between MrBayes and PhyloTextDiff. Each point is a cluster, colored by mean unnormalized log-posterior. Points above the red dashed line indicate clusters better represented by PhyloTextDiff. Counts are normalized, and $\epsilon = 10^{-5}$ is added for log–log plotting.

We further analyze the tree space explored by MrBayes and PhyloTextDiff by randomly sampling 2,500 trees from each model and computing pairwise Robinson–Foulds (RF) distances (Robinson and Foulds, 1981), which quantify topological differences between trees based on differing bipartitions. Using the resulting RF distance matrix, we perform hierarchical clustering with a threshold set at 30% of the maximum RF distance; the remaining trees are assigned to the nearest cluster medoid. This procedure identifies a total of 76 clusters. Figure 5 presents the results. Each point represents a cluster, with the color indicating the mean unnormalized posterior probability of that cluster. Counts are normalized by the total number of trees for each method. A small $\epsilon = 10^{-4}$ was added to include clusters with zero trees on the log–log scale (gray lines). The red dashed line corresponds to $y = x$; points above this line indicate clusters where PhyloTextDiff contributes more trees than MrBayes, while points below indicate the opposite. We observe that for both methods, clusters with higher unnormalized mean posterior (top right) have higher sampling frequencies. However, clusters with lower unnormalized mean posterior (middle) are better represented by PhyloTextDiff. On the two gray lines, we see that PhyloTextDiff successfully discovers eight clusters and fails to recover only one. A closer analysis of these nine clusters is provided in Appendix J.4 and another posterior analysis with equal sampling in Appendix J.3. We also discuss runtime considerations in Appendix J.5.

## 6 SUMMARY AND CONCLUDING REMARKS

We introduced PhyloTextDiff, the first discrete diffusion model for phylogenetic inference that operates directly on a text-based representation of tree topologies. Our framework integrates a custom tokenizer, DNA-informed token embeddings, a conditioned loss and architecture, and an efficient sampling strategy. Evaluations on eight real datasets, each trained under three different phylogenetic models, show that PhyloTextDiff outperforms baselines in terms of MLL and ELBO, while efficiently exploring tree space. Notably, the diffusion model achieves comprehensive results for all datasets within only six hours of training. Looking ahead, we aim to extend PhyloTextDiff toward more generalizable and end-to-end phylogenetic inference. A key direction is enabling zero-shot and few-shot inference, so that the model can seamlessly handle unseen taxa or datasets without retraining. This will require rethinking the tokenization scheme. Another important avenue is the joint modeling of branch lengths and topologies, for example by coupling discrete diffusion on tree structures with conditional models that generate both simultaneously. Beyond DNA sequences, broadening the framework to proteins or morphological traits through specialized encoders would further expand its applicability. Together, these extensions could move the field closer to scalable, foundation-style generative models for phylogenetics. For a more detailed discussion, see Appendix L.

## REPRODUCIBILITY

Code and data will be made available. See Appendix M for more details.

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

# A  PHYLOGENETIC LIKELIHOOD AND THE PRUNING ALGORITHM

The evolutionary process along a phylogenetic tree is modeled as a Markov process on characters (Felsenstein, 2004). Let $\Sigma$ denote the alphabet of possible character states (e.g., nucleotides). For site $i \in \{1, \ldots, m\}$ and node $v \in \mathcal{V}$, let $a_v^i \in \Sigma$ denote the character assigned to node $v$. The set of leaf nodes is denoted by $\mathcal{L}$, and for $v \in \mathcal{L}$ the assignments $a_v^i$ are observed from the sequence alignment $\mathbf{Y}$.

We denote the root by $r$. For each site $i$, the root state $a_r^i$ is assumed to be drawn independently from the stationary distribution $\eta(\cdot)$ of the substitution model.

For any edge $(u, v) \in \mathcal{E}(\tau)$ with branch length $b_{uv}$, the transition probability from state $a_u^i$ at node $u$ to state $a_v^i$ at node $v$ is given by $P_{a_u^i a_v^i}(b_{uv})$, where $P(\cdot)$ is determined by a continuous-time substitution model (e.g., Jukes and Cantor, 1969).

Assuming independence across sites, the likelihood of observing $\mathbf{Y}$ given topology $\tau$ and branch lengths $\mathbf{b}$ is

$$P(\mathbf{Y} \mid \tau, \mathbf{b}) = \prod_{i=1}^{m} P(Y_i \mid \tau, \mathbf{b}) = \prod_{i=1}^{m} \sum_{a_i} \left( \eta(a_r^i) \prod_{(u,v) \in \mathcal{E}(\tau)} P_{a_u^i a_v^i}(b_{uv}) \right), \qquad (9)$$

where $a_i$ ranges over all extensions of $Y_i$ to the internal nodes, $\mathcal{E}(\tau)$ is the set of edges of $\tau$, and the leaf assignments $a_v^i$ for $v \in \mathcal{L}$ are fixed by the data.

The likelihood can be evaluated efficiently using Felsenstein's pruning algorithm (Felsenstein, 2004). This is a bottom-up dynamic programming procedure performed as a post-order traversal of the tree.

Let $L_u^i$ denote the sequences at site $i$ in the subtree below an internal node $u$, and let $v$ and $w$ be its two children. The conditional probability of observing $L_u^i$ given that node $u$ is in state $a_u^i$ is

$$P(L_u^i \mid a_u^i) = \sum_{a_v^i, a_w^i \in \Sigma} P(a_v^i \mid a_u^i, b(e_v)) \, P(L_v^i \mid a_v^i) \, P(a_w^i \mid a_u^i, b(e_w)) \, P(L_w^i \mid a_w^i), \qquad (10)$$

where $b(e_v)$ and $b(e_w)$ are the branch lengths of edges connecting $u$ to $v$ and $w$, respectively.

At the leaves, the conditional probabilities are represented using one-hot encoding of the observed character, so that for leaf node $l$:

$$P(L_l^i \mid a_l^i = c) = \begin{cases} 1, & \text{if the observed character at site } i \text{ is } c, \\ 0, & \text{otherwise.} \end{cases} \qquad (11)$$

Finally, the likelihood at site $i$ is obtained at the root node $r$ as

$$P(Y_i \mid \tau, \mathbf{b}) = \sum_{a_r^i \in \Sigma} \eta(a_r^i) \, P(L_r^i \mid a_r^i), \qquad (12)$$

where $\eta(a_r^i)$ is the stationary distribution of the substitution model at the root. The full likelihood for the alignment is then

$$P(\mathbf{Y} \mid \tau, \mathbf{b}) = \prod_{i=1}^{m} P(Y_i \mid \tau, \mathbf{b}). \qquad (13)$$

Finally, by the *Pulley principle* (Felsenstein, 1981), the placement of the root does not affect the computed likelihood.

# B  SCORE-BASED DIFFUSION MODELS

Score-based diffusion models originate from Langevin dynamics, designed to sample from a probability density $p(x)$ using only gradients $\nabla_x \log p(x)$. In continuous space, Stochastic Gradient Langevin Dynamics (SGLD) iteratively updates:

$$x_t = x_{t-1} + \frac{\delta}{2} \nabla_x \log p(x_{t-1}) + \sqrt{\delta} \, \epsilon_t, \quad \epsilon_t \sim \mathcal{N}(0, I). \qquad (14)$$

A *score network* $s_\theta : \mathbb{R}^D \to \mathbb{R}^D$ learns the score $\nabla_x \log p(x)$ by minimizing the objective

$$\mathbb{E}_{p_{\text{data}}}\left[\|s_\theta(x) - \nabla_x \log p_{\text{data}}(x)\|^2\right], \tag{15}$$

with scalable variants including *denoising score matching* (Vincent, 2011) and *sliced score matching* (Song et al., 2019).

**Discrete diffusion.** For discrete spaces $\mathcal{X} = \{1, \ldots, K\}$, the forward process is defined via a transition matrix $Q_t \in [0,1]^{K \times K}$:

$$[Q_t]_{ij} = q(x_t = j \mid x_{t-1} = i), \quad q(x_t \mid x_0) = \text{Cat}(x_t; p = x_0 \prod_{i=1}^t Q_i), \tag{16}$$

where $Cat$ denotes a categorical distribution. The rows of $Q_t$ sum to one and $\prod_{i=1}^t Q_i$ converges to a stationary distribution.

Inspired by masked language models, $Q_t$ can include an *absorbing [MASK] state* $m$:

$$Q_t = (1 - \beta_t)I + \beta_t \mathbf{1}_{e_m^T}, \quad [Q_t]_{ij} = \begin{cases} 1 & i = j = m \\ 1 - \beta_t & i = j \neq m \\ \beta_t & i \neq m, j = m \end{cases}. \tag{17}$$

**Continuous-time framework.** Campbell et al. (2022) define a transition *rate matrix* $R_t$ with

$$q_{t|t-\Delta t}(x \mid \hat{x}) = \delta_{x,\hat{x}} + R_t(\hat{x}, x)\Delta t + o(\Delta t), \tag{18}$$

allowing the forward distribution to be computed analytically as

$$q_{t|0}(x \mid x_0) = \left[\exp\left(\int_0^t R_s ds\right)\right]_{x_0, x}. \tag{19}$$

For high-dimensional spaces, each dimension can be propagated independently due to factorization of the forward process.

**Reverse-time process.** The generative reverse-time process $\bar{X}_t = X_{T-t}$ is also Markovian (Sun et al., 2023), with transition probabilities

$$q_{s|t}(x_s \mid x_t) = \frac{q_s(x_s)}{q_t(x_t)} q_{t|s}(x_t \mid x_s), \quad s < t, \tag{20}$$

implying a backward rate matrix

$$\bar{R}_t(x, y) = \frac{q_t(y)}{q_t(x)} R_t(y, x). \tag{21}$$

Learning this ratio is intractable jointly, so it is approximated dimension by dimension using a neural network.

**Diffusion-weighted denoising score entropy (DWDSE) loss.** Lou et al. (2024) defined a loss for learning the score $s_\theta(x, t)$:

$$\mathcal{L}_{\text{DWDSE}}(x_0) = \int_0^T \mathbb{E}_{x_t \sim q_{t|0}(\cdot|x_0)} \sum_{y \neq x_t} R_t(x_t, y)\left[s_\theta(x_t, t)_y - \frac{q_{t|0}(y|x_0)}{q_{t|0}(x_t|x_0)} \log s_\theta(x_t, t)_y + K\left(\frac{q_{t|0}(y|x_0)}{q_{t|0}(x_t|x_0)}\right)\right] dt, \tag{22}$$

with $K(a) = a(\log a - 1)$. This upper-bounds the negative log-likelihood:

$$-\log q_0^\theta(x_0) \leq \mathcal{L}_{\text{DWDSE}}(x_0) + D_{\text{KL}}(q_{T|0}(\cdot|x_0) \| p_{\text{base}}). \tag{23}$$

## C  Phylogenetic Tokenizer and Textual Representation of Phylogenetic Trees

The position of the root does not impact the likelihood of a tree, so we unroot all the trees. Each textual tree topology from dataset $j$ with $n_j$ taxa has then a fixed length of $4n_j - 4$ words. We note $d_j = 4n_j - 4$ the length of the textual tree topologies from dataset $j$ and $d = \max_{1 \leq j \leq N} d_j$ the maximum length of trees across all the datasets.

With $\mathcal{S} = \{\texttt{<PAD>}, \texttt{<UNK>}, \texttt{<SOS>}, \texttt{<EOS>}\}$, the special tokens set, $\mathcal{N} = \{\, (\,,\,)\,,\,,\,,\,;\,\}$, the Newick format characters set, and $\mathcal{T}_j = \{t_0, \cdots, t_{n_j}\}$ the taxa names set, we define $\mathcal{W}_j$, the Newick vocabulary for a dataset $j$ with $n_j$ taxa, as $\mathcal{W}_j = \mathcal{T}_j \cup \mathcal{N} \cup \mathcal{S}$. For a collection of $N$ datasets, with $\mathcal{T}_{1:N} = \bigcup_{j=1}^{N} \mathcal{T}_j$, we define the Newick vocabulary as $\mathcal{W}_{1:N} = \mathcal{T}_{1:N} \cup \mathcal{N} \cup \mathcal{S}$. We design a custom tokenizer that maps our textual tree topologies $\tau_j \in \mathcal{W}_j^{d_j}$ to vectors $\boldsymbol{x} \in \mathcal{X} = [0, \cdots, |\mathcal{W}_{1:N}| - 1]^{d+2}$:

$$\text{Tokenizer}: \quad \begin{aligned} [1, \cdots, N] \times \bigcup_{j=1}^{N} \mathcal{W}_j^{d_j} &\longrightarrow \mathcal{X} = [0, \cdots, |\mathcal{W}_{1:N}| - 1]^{d+2} \\ j, \tau_j = (\tau_j^{(0)}, \cdots, \tau_j^{(d_j - 1)}) &\longmapsto \boldsymbol{x} = [\texttt{<SOS>}, x^{(0)}, \cdots, x^{(d-1)}, \texttt{<EOS>}], \end{aligned} \quad (24)$$

where the target space $\mathcal{X} = [0, \cdots, |\mathcal{W}_{1:N}| - 1]^{d+2}$ is the set of $(d+2)$-dimensional vectors with values in $\{0, \cdots, |\mathcal{W}_{1:N}| - 1\}$, and $|\mathcal{W}_{1:N}|$ denotes the cardinality of the vocabulary set $\mathcal{W}_{1:N}$. We design our tokenizer so that the first 4 tokens are assigned to $\mathcal{S}$, and the next 4 tokens to $\mathcal{N}$. The remaining $|\mathcal{W}_{1:N}| - 8$ tokens are then assigned to $\mathcal{T}_{1:N}$. Each sentence $\tau_j \in \mathcal{W}_j^{d_j}$ from dataset $j$ is constructed by first placing the $\texttt{<SOS>}$ token, followed by the $4n_j - 4$ tokens corresponding to elements from $\mathcal{N} \cup \mathcal{T}_j$, the $\texttt{<EOS>}$ token, and finally padding tokens to fill the remaining $d - d_j$ positions. Our tokenizer is both fast and scalable. Unlike standard approaches that require learning a tokenization scheme (e.g., subword segmentation), ours treats taxa names as predefined indices. This eliminates the need for training and results in a small, fixed vocabulary, which reduces the computational cost of the downstream neural network, as its matrix sizes depend on vocabulary size.

## D  Marginal Likelihood Estimation

We aim to minimize the KL between the true and the learned posterior distribution:

$$\theta^*, \phi^* = \arg\min_{\phi, \psi} \mathrm{D}_{\mathrm{KL}} \left( q_{\theta,\phi}(\tau, \mathbf{b}) \,\|\, p(\tau, \mathbf{b} \mid \boldsymbol{Y}_j) \right), \quad (25)$$

which is equivalent to maximizing the following evidence lower bound:

$$\mathbb{E}_{q_{\theta,\phi}(\tau, \mathbf{b})} \log \left( \frac{P(\boldsymbol{Y}_j | \tau, \mathbf{b}) p(\tau, \mathbf{b})}{q_\theta(\tau) q_\phi(\mathbf{b} | \tau)} \right) \leq \log p(\boldsymbol{Y}_j). \quad (26)$$

However, we cannot evaluate $q_\theta(\tau)$, so we instead evaluate this evidence lower bound:

$$\mathbb{E}_{\substack{\boldsymbol{x}_{0:T} \sim q_{R,j} \\ \boldsymbol{b} \sim q_\phi(\boldsymbol{b} | \boldsymbol{x}_0)}} \log \left( p(\boldsymbol{Y}_j | \boldsymbol{x}_0, \boldsymbol{b}) p(\boldsymbol{x}_0) \frac{q_F(\boldsymbol{x}_{1:T} | \boldsymbol{x}_0)}{q_{R,j}(\boldsymbol{x}_{0:T})} \frac{p(\boldsymbol{b} | \boldsymbol{x}_0)}{q_\phi(\boldsymbol{b} | \boldsymbol{x}_0)} \right) \leq \log p(\boldsymbol{Y}_j). \quad (27)$$

**Proof.** Let $\boldsymbol{x}_0 \sim p_{data}$. We define the trajectory $\boldsymbol{x}_{0:T} = (\boldsymbol{x}_0, \boldsymbol{x}_1, \ldots, \boldsymbol{x}_T)$. We note the forward process $q_F(\boldsymbol{x}_{0:T}) = p_{\text{data}}(\boldsymbol{x}_0) \prod_{t=1}^{T} q_{t|t-1}(\boldsymbol{x}_t | \boldsymbol{x}_{t-1})$ and the reverse process $q_{R,j}(\boldsymbol{x}_{0:T}) = p_{\text{base}}(\boldsymbol{x}_T) \prod_{t=1}^{T} q_{t-1|t}(\boldsymbol{x}_{t-1} | \boldsymbol{x}_t, j)$.

Now let's consider $q_F(\boldsymbol{x}_{1:T} | \boldsymbol{x}_0) = \prod_{t=1}^{T} q_{t-1|t}(\boldsymbol{x}_{t-1} | \boldsymbol{x}_t)$, which is the probability of generating a trajectory $\boldsymbol{x}_1, \ldots, \boldsymbol{x}_T$ given initial state $\boldsymbol{x}_0$. This is a distribution over all the trajectories and therefore it sums to one:

$$\sum_{\boldsymbol{x}_{1:T}} q_F(\boldsymbol{x}_{1:T} | \boldsymbol{x}_0) = \sum_{\boldsymbol{x}_1} \sum_{\boldsymbol{x}_2} \cdots \sum_{\boldsymbol{x}_T} q_{1|0}(\boldsymbol{x}_1 | \boldsymbol{x}_0) q_{2|1}(\boldsymbol{x}_2 | \boldsymbol{x}_1) \cdots q_{T|T-1}(\boldsymbol{x}_T | \boldsymbol{x}_{T-1}) = 1 \quad (28)$$

We can thus derive the following:

$$P(\mathbf{Y}) = \int_{\boldsymbol{b}} \sum_{\tau \in \mathcal{X}} P(\mathbf{Y}|\tau, \boldsymbol{b}) p(\tau) p(\boldsymbol{b}|\tau) d\boldsymbol{b} \tag{29}$$

$$= \int_{\boldsymbol{b}} \sum_{\boldsymbol{x}_0 \in \mathcal{X}} P(\mathbf{Y}|\boldsymbol{x}_0, \boldsymbol{b}) p(\boldsymbol{x}_0) p(\boldsymbol{b}|\boldsymbol{x}_0) d\boldsymbol{b} \times \sum_{\boldsymbol{x}_{1:T}} q_F(\boldsymbol{x}_{1:T}|\boldsymbol{x}_0) \tag{30}$$

$$= \int_{\boldsymbol{b}} \sum_{\boldsymbol{x}_0} \sum_{\boldsymbol{x}_{1:T}} P(\mathbf{Y}|\boldsymbol{x}_0, \boldsymbol{b}) p(\boldsymbol{x}_0) p(\boldsymbol{b}|\boldsymbol{x}_0) q_F(\boldsymbol{x}_{1:T}|\boldsymbol{x}_0) d\boldsymbol{b} \tag{31}$$

$$= \int_{\boldsymbol{b}} \sum_{\boldsymbol{x}_{0:T}} q_{R,j}(\boldsymbol{x}_{0:T}) q_\phi(\boldsymbol{b}|\boldsymbol{x}_0) \frac{P(\mathbf{Y}|\boldsymbol{x}_0, \boldsymbol{b}) p(\boldsymbol{x}_0) p(\boldsymbol{b}|\boldsymbol{x}_0) q_F(\boldsymbol{x}_{1:T}|\boldsymbol{x}_0)}{q_{R,j}(\boldsymbol{x}_{0:T}) q_\phi(\boldsymbol{b}|\boldsymbol{x}_0)} d\boldsymbol{b} \tag{32}$$

$$= \sum_{\boldsymbol{x}_{0:T}} q_{R,j}(\boldsymbol{x}_{0:T}) \int_{\boldsymbol{b}} q_\phi(\boldsymbol{b}|\boldsymbol{x}_0) P(\mathbf{Y}|\boldsymbol{x}_0, \boldsymbol{b}) p(\boldsymbol{x}_0) \frac{q_F(\boldsymbol{x}_{1:T}|\boldsymbol{x}_0)}{q_{R,j}(\boldsymbol{x}_{0:T})} \frac{p(\boldsymbol{b}|\boldsymbol{x}_0)}{q_\phi(\boldsymbol{b}|\boldsymbol{x}_0)} d\boldsymbol{b} \tag{33}$$

$$= \mathbb{E}_{\boldsymbol{x}_{0:T} \sim q_{R,j}} \mathbb{E}_{\boldsymbol{b} \sim q_\phi(\boldsymbol{b}|\boldsymbol{x}_0)} P(\mathbf{Y}|\boldsymbol{x}_0, \boldsymbol{b}) p(\boldsymbol{x}_0) \frac{q_F(\boldsymbol{x}_{1:T}|\boldsymbol{x}_0)}{q_{R,j}(\boldsymbol{x}_{0:T})} \frac{p(\boldsymbol{b}|\boldsymbol{x}_0)}{q_\phi(\boldsymbol{b}|\boldsymbol{x}_0)}. \tag{34}$$

$$\tag{35}$$

By sampling $K$ i.i.d. pairs $(\boldsymbol{x}_{0:T}, \boldsymbol{b}) \sim q_{R,j} q_\phi$, let the importance weights be defined as

$$w(\boldsymbol{x}_{0:T}, \boldsymbol{b}) = p(\mathbf{Y}|\boldsymbol{x}_0, \boldsymbol{b}) p(\boldsymbol{x}_0) \frac{q_F(\boldsymbol{x}_{1:T}|\boldsymbol{x}_0)}{q_{R,j}(\boldsymbol{x}_{0:T})} \frac{p(\boldsymbol{b}|\boldsymbol{x}_0)}{q_\phi(\boldsymbol{b}|\boldsymbol{x}_0)}. \tag{36}$$

We define the following random variable:

$$\hat{Z} = \frac{1}{K} \sum_{\substack{i=1 \\ \boldsymbol{x}_{0:T} \sim q_{R,j} \\ \boldsymbol{b} \sim q_\phi(\boldsymbol{b}|\boldsymbol{x}_0)}}^{K} w(\boldsymbol{x}_{0:T}, \boldsymbol{b}). \tag{37}$$

Thus,

$$\mathbb{E}_{\substack{\boldsymbol{x}_{0:T} \sim q_{R,j} \\ \boldsymbol{b} \sim q_\phi(\boldsymbol{b}|\boldsymbol{x}_0)}} [\hat{Z}] = \mathbb{E}_{\substack{\boldsymbol{x}_{0:T} \sim q_{R,j} \\ \boldsymbol{b} \sim q_\phi(\boldsymbol{b}|\boldsymbol{x}_0)}} \frac{1}{K} \sum_{\substack{i=1 \\ \boldsymbol{x}_{0:T} \sim q_{R,j} \\ \boldsymbol{b} \sim q_\phi(\boldsymbol{b}|\boldsymbol{x}_0)}}^{K} w(\boldsymbol{x}_{0:T}, \boldsymbol{b}) = P(\mathbf{Y}). \tag{38}$$

By Jensen's inequality applied to the concave function $\log(\cdot)$, we have:

$$\mathbb{E}_{\substack{\boldsymbol{x}_{0:T} \sim q_{R,j} \\ \boldsymbol{b} \sim q_\phi(\boldsymbol{b}|\boldsymbol{x}_0)}} [\log \hat{Z}] \leq \log \mathbb{E}_{\substack{\boldsymbol{x}_{0:T} \sim q_{R,j} \\ \boldsymbol{b} \sim q_\phi(\boldsymbol{b}|\boldsymbol{x}_0)}} [\hat{Z}] = \log P(\mathbf{Y}), \tag{39}$$

i.e.

$$\mathbb{E}_{\substack{\boldsymbol{x}_{0:T} \sim q_{R,j} \\ \boldsymbol{b} \sim q_\phi(\boldsymbol{b}|\boldsymbol{x}_0)}} \log \left( \frac{1}{K} \sum_{\substack{i=1 \\ \boldsymbol{x}_{0:T} \sim q_{R,j} \\ \boldsymbol{b} \sim q_\phi(\boldsymbol{b}|\boldsymbol{x}_0)}}^{K} P(\mathbf{Y}|\boldsymbol{x}_0, \boldsymbol{b}) p(\boldsymbol{x}_0) \frac{q_F(\boldsymbol{x}_{1:T}|\boldsymbol{x}_0)}{q_{R,j}(\boldsymbol{x}_{0:T})} \frac{p(\boldsymbol{b}|\boldsymbol{x}_0)}{q_\phi(\boldsymbol{b}|\boldsymbol{x}_0)} \right) \leq \log P(\mathbf{Y}). \tag{40}$$

## E  BRANCH LENGTH MODEL

We adopt the strategy introduced in VBPI-GNN (Zhang, 2023). Our exposition here follows the original paper closely, and we refer readers to it for full details. Following this approach, we assume that node features vary smoothly across the tree topology - that is, the feature vector of each node is similar to those of its neighbors. A common measure of smoothness for functions defined on the nodes of a graph is the *Dirichlet energy*.

Formally, given a tree topology $\tau = (V, E)$, where $V$ denotes the set of nodes and $E$ the set of edges, and a feature mapping $f : V \to \mathbb{R}^d$, the Dirichlet energy is defined as:

$$l(f, \tau) = \sum_{(u,v) \in E} \| f(u) - f(v) \|^2. \tag{41}$$

Let $V = V^b \cup V^\circ$, where $V^b$ denotes the set of leaf nodes and $V^\circ$ denotes the set of interior nodes. Let $X^b = \{x_v | v \in V^b\}$ be the set of one-hot embeddings for the leaf nodes. The interior node features $X^\circ = \{x_v | v \in V^\circ\}$ then can be obtained by minimizing the Dirichlet energy

$$\hat{X}^\circ = \arg\min_{X^\circ} \ell(X^\circ, X^b, \tau) = \arg\min_{X^\circ} \sum_{(u,v) \in E} \| x_u - x_v \|^2. \tag{42}$$

Note that the above Dirichlet energy function is convex, its minimizer therefore can be obtained by solving the following optimality condition

$$\frac{\partial \ell(\mathbf{X}^o, \mathbf{X}^b, \tau)}{\partial \mathbf{X}^o}(\widehat{\mathbf{X}}^o) = \mathbf{0}. \tag{43}$$

It turns out that this equation has a close-form solution based on matrix inversion. However, as matrix inversion scales cubically in general, it is infeasible for graphs with many nodes. Fortunately, by leveraging the hierarchical structure of phylogenetic trees, we can design a more efficient linear time algorithm as follows. We first rewrite the equation as a system of linear equations

$$\sum_{v \in \mathcal{N}(u)} (\widehat{x}_u - \widehat{x}_v) = 0, \quad \forall u \in V^o, \quad \widehat{x}_v = x_v, \quad \forall v \in V^b, \tag{44}$$

where $\mathcal{N}(u)$ is the set of neighbors of node $u$. Given a topological ordering induced by the tree, we can obtain the solution within a two-pass sweep through the tree topology, similar to the Thomas algorithm (Thomas, 1949) for solving tridiagonal systems of linear equations. In the first pass, we traverse the tree in a postorder fashion and express the node features as a linear function of those of their parents,

$$\widehat{x}_u = c_u \widehat{x}_{\pi_u} + d_u, \tag{45}$$

for all the nodes except the root node, where $\pi_u$ denotes the parent node of $u$. More specifically, we first initialize $c_u = 0$, $d_u = x_u$ for all leaf nodes $u \in V^b$. For all the interior nodes except the root node, we compute $c_u, d_u$ recursively as follows:

$$c_u = \frac{1}{|\mathcal{N}(u)| - \sum_{v \in \text{ch}(u)} c_v}, \quad d_u = \frac{\sum_{v \in \text{ch}(u)} d_v}{|\mathcal{N}(u)| - \sum_{v \in \text{ch}(u)} c_v}, \tag{46}$$

where $\text{ch}(u)$ denotes the set of child nodes of $u$. In the second pass, we traverse the tree in a preorder fashion and compute the solution by back substitution. Concretely, at the root node $r$, we can compute the node feature directly as below:

$$\widehat{x}_r = \frac{\sum_{v \in \text{ch}(r)} d_v}{|\mathcal{N}(r)| - \sum_{v \in \text{ch}(r)} c_v}. \tag{47}$$

For all the other interior nodes, the node features can be obtained by substituting the learned features for the parent nodes. Moreover, the algorithm is proven to be numerically stable.

Once we have the node features, we can derive the branch features using a graph neural network:

$$\forall e = (u, v) \in E, \quad h_e = \text{GNN}(h_u, h_v) \tag{48}$$

and finally, by chosing the branch length to follow a diagonal lognormal distribution, we can derive the mean and standard variation parameters :

$$\mu(e, \tau) = \text{MLP}^\mu(h_e), \quad \sigma(e, \tau) = \text{MLP}^\sigma(h_e) \tag{49}$$

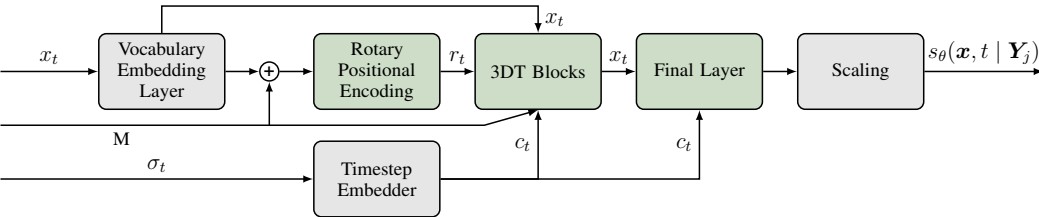

Figure 6: Detailed Architecture of PhyloTextDiff. Input tokens $x_t \sim q_{t|0}(\mathbf{x}_0)$ are first embedded using a learnable embedding matrix and enriched with DNA embeddings $\mathbf{M}$. These embeddings are combined with Rotary Positional Encodings and timestep embeddings $\sigma_t$, then processed through 3DT blocks with cross-attention on $\mathbf{M}$. The final output is the predicted score $s_\theta(\boldsymbol{x}, t \mid \boldsymbol{Y}_j)$.

## F ARCHITECTURE

Our architecture is described in Fig.6.

We first begin by sampling from the forward distribution:

$$\mathbf{x}_t \sim q_{t|0}(\mathbf{x}_t|\mathbf{x}_0) = \left(\exp\left[\int_0^t \mathbf{R}_s \ ds\right]\right)_{x_0}, t \sim \mathcal{U}(0, 1) \tag{50}$$

To embed our trees, we combine semantic and structural information of our tokens with genetic information. Let $\mathbf{x}_t = (x_t^{(1)}, \ldots, x_t^{(d)})$ be a sequence of tokens, and let $\mathbf{E} \in \mathbb{R}^{(|\mathcal{W}|+1) \times h}$ be a learnable embedding matrix. Specifically, each token $x_t^{(i)}$ is embedded as the sum of its learnable embedding $\mathbf{E}_{x_t^{(i)}}$ and the corresponding row of $\mathbf{M}$, $\mathbf{M}_{x_t^{(i)}}$, i.e., $\mathbf{h}_t^{(i)} = \mathbf{E}_{x_t^{(i)}} + \mathbf{M}_{x_t^{(i)}}$.

A Rotary Positional Encoding (RoPE) is then applied :

$$\mathbf{r}_t = RoPE(\mathbf{h}_t) \tag{51}$$

The Timestep Embedder is inspired by the approach described in the Glide framework (Nichol et al., 2022). A sinusoidal embedding is computed based on the noise levet $\sigma_t$ using exponentially spaced frequencies and then passed trough an MLP and a Sigmoid Linear Unit (SiLU) function

$$\mathbf{c}_t = \text{SiLU}(\text{Timestep\_Embedder}(\sigma_t)). \tag{52}$$

We design the DNA Discrete Diffusion Transformer (3DT) blocks to enable dataset-aware guided diffusion. Specifically, we introduce a Multi-Head Cross-Attention block in each transformer layer, where the guidance source is the previously defined matrix $\mathbf{M}$. The architecture of the 3DT block is described in Fig. 7.

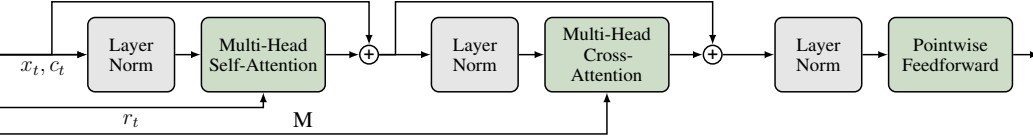

Figure 7: Architecture of a single 3DT block in PhyloTextDiff. The block receives token embeddings $x_t$ and timestep embeddings $c_t$, along with positional encodings $r_t$. Inputs are first normalized, then processed through Multi-Head Self-Attention and a residual connection, followed by Layer Norm and Multi-Head Cross-Attention with guidance from $\mathbf{M}$. A final Layer Norm and pointwise feedforward layer produce the block output.

Finally, we use a final layer composed of a layer normalization and of a linear layer and scale the output by $\sigma_t$. The output of the model is the concrete score $s_\theta(\mathbf{x}, t \mid \boldsymbol{Y}_j)$.

## G    DATASET INFORMATION

Table 4 summarizes the benchmark datasets (DS1–DS8) used in our experiments, including the number of species, alignment length in sites, and references. Each dataset corresponds to a multiple sequence alignment of DNA sequences, where sites represent aligned nucleotide ($\{A, C, G, T\}$) positions.

Table 4: Statistics of the benchmark datasets from DS1 to DS8.

| Dataset | # Species | # Sites | Reference |
|---------|-----------|---------|-----------|
| DS1 | 27 | 1949 | (Hedges et al., 1990) |
| DS2 | 29 | 2520 | (Garey et al., 1996) |
| DS3 | 36 | 1812 | (Yang and Yoder, 2003) |
| DS4 | 41 | 1137 | (Henk et al., 2003) |
| DS5 | 50 | 378 | (Lakner et al., 2008) |
| DS6 | 50 | 1133 | (Zhang and Blackwell, 2001) |
| DS7 | 59 | 1824 | (Yoder and Yang, 2004) |
| DS8 | 64 | 1008 | (Rossman et al., 2001) |

## H    TRAINING DETAILS

We adopt the same hyperparameter settings as Lou et al. (2024) and train our model on 2 A100 GPUs with 64 GB of memory.

Table 5: Hyperparameters and training details for PhyloTextDiff.

| Category | Value |
|----------|-------|
| **Model Architecture** | |
| DDiT Hidden size | 768 |
| Timestep embedding dim | 128 |
| Number of transformer blocks | 12 (small), 24 (medium) |
| Number of attention heads | 12 (small), 16 (medium) |
| Dropout | 0.1 |
| Scale by $\sigma$ | True |
| **Training** | |
| Batch size | 64 |
| Gradient accumulation | 1 |
| Number of training steps | 80,000 (small), 90,000 (medium) |
| Learning rate | 3e-4 |
| Optimizer | AdamW ($\beta_1 = 0.9$, $\beta_2 = 0.999$, $\epsilon = 1e-8$) |
| Weight decay | 0 |
| Warmup steps | 2,500 |
| Gradient clipping | 1.0 |
| EMA | 0.9999 |
| **Noise / Forward Process** | |
| Type | loglinear |
| $\sigma_{\min}$ | 1e-4 |
| $\sigma_{\max}$ | 20 |
| $\epsilon$ | 1e-3 |

## I    DETAILS ON MARGINAL LOG-LIKELIHOOD (MLL) ESTIMATION

### I.1    RESULTS FROM OTHER METHODS

The MLL results for the baseline are taken from (Duan et al., 2024) except for SBN (Zhang and Iv, 2019) which is taken from from (Xie and Zhang, 2023). We experimentally veried the results for MrBayes (Ronquist et al., 2012), VBPI-GNN (Zhang, 2023) and PhyloGFN (Zhou et al., 2024a).

## I.2 Lower Bound computations for other methods

### I.2.1 MrBayes SS

We use MrBayes (Ronquist et al., 2012) with the stepping-stone (SS) method (Xie et al., 2011) to derive its MLL estimation. The marginal likelihood of a dataset $D$ under model $M$ is

$$p(D \mid M) = \int p(D \mid \theta, M) \, p(\theta \mid M) \, d\theta, \tag{53}$$

where $\theta$ represents all model parameters, including tree topology and substitution model parameters. Direct computation of this integral is intractable.

Stepping-stone sampling introduces a series of power posteriors defined by a temperature parameter $\beta \in [0, 1]$:

$$p_\beta(\theta \mid D, M) \propto p(D \mid \theta, M)^\beta \, p(\theta \mid M). \tag{54}$$

Let $0 = \beta_0 < \beta_1 < \cdots < \beta_K = 1$ denote the sequence of stepping stones. The marginal likelihood can be written as a telescoping product:

$$p(D \mid M) = \prod_{k=1}^{K} \frac{Z_{\beta_k}}{Z_{\beta_{k-1}}}, \tag{55}$$

where $Z_{\beta_k}$ is the normalizing constant of the $\beta_k$-th power posterior.

Each ratio is approximated using MCMC samples $\{\theta_i^{(k)}\}_{i=1}^{n_k}$ from $p_{\beta_k}(\theta \mid D, M)$:

$$\frac{Z_{\beta_k}}{Z_{\beta_{k-1}}} \approx \frac{1}{n_k} \sum_{i=1}^{n_k} p(D \mid \theta_i^{(k)}, M)^{\beta_k - \beta_{k-1}}. \tag{56}$$

Hence, the log marginal likelihood is estimated as

$$\log p(D \mid M) \approx \sum_{k=1}^{K} \log \left( \frac{1}{n_k} \sum_{i=1}^{n_k} p(D \mid \theta_i^{(k)}, M)^{\beta_k - \beta_{k-1}} \right). \tag{57}$$

### I.2.2 SBN (Zhang and Iv, 2019), VBPI-GNN (Zhang, 2023), GeoPhy (Mimori and Hamada, 2023), ARTree (Xie and Zhang, 2023)

The variational approximation is factorized as

$$Q_{\phi,\psi}(\tau, q) = Q_\phi(\tau) \, Q_\psi(q \mid \tau). \tag{58}$$

The KL divergence to the true posterior is minimized as

$$(\phi^*, \psi^*) = \arg\min_{\phi,\psi} D_{\mathrm{KL}}\big(Q_{\phi,\psi}(\tau, q) \,\|\, p(\tau, q \mid Y)\big). \tag{59}$$

The Evidence Lower Bound (ELBO) is

$$\mathcal{L}(\phi, \psi) = \mathbb{E}_{Q_{\phi,\psi}(\tau,q)}\left[ \log \frac{p(Y \mid \tau, q) \, p(\tau, q)}{Q_\phi(\tau) \, Q_\psi(q \mid \tau)} \right] \leq \log p(Y). \tag{60}$$

The multi-sample ELBO is

$$\mathcal{L}_K(\phi, \psi) = \mathbb{E}_{Q_{\phi,\psi}(\tau^{1:K}, q^{1:K})}\left[ \log \frac{1}{K} \sum_{i=1}^{K} \frac{p(Y \mid \tau^i, q^i) \, p(\tau^i, q^i)}{Q_\phi(\tau^i) \, Q_\psi(q^i \mid \tau^i)} \right] \leq \log p(Y), \tag{61}$$

where

$$Q_{\phi,\psi}(\tau^{1:K}, q^{1:K}) = \prod_{i=1}^{K} Q_{\phi,\psi}(\tau^i, q^i). \tag{62}$$

### I.2.3 VAIPHY (KOPTAGEL ET AL., 2023)

Vaiphy performs variational inference in an augmented space $\mathcal{A}$ by approximating the posterior

$$p_\theta(\tau, B, Z \mid X) \propto p_\theta(X, Z \mid B, \tau)\, p_\theta(B \mid \tau)\, p_\theta(\tau) \tag{63}$$

with a factorized variational distribution

$$q(\tau, B, Z \mid X) = q(B \mid X)\, q(\tau \mid X)\, q(Z \mid X), \quad q(Z \mid X) = \prod_{i \in \mathcal{I}(\mathcal{A})} q(Z_i \mid X), \tag{64}$$

where $\mathcal{I}(\mathcal{A})$ is the set of internal vertices in the augmented space.

The expected number of mutations along edge $(i, j)$ is computed as

$$\phi_{ij} \in [0, M], \tag{65}$$

where $M$ is the number of sites.

The framework learns its parameters by maximizing the ELBO:

$$\mathcal{L} = \mathbb{E}_{q(\tau, B, Z \mid X)}\left[ \log \frac{p_\theta(X, Z \mid B, \tau)\, p_\theta(B \mid \tau)\, p_\theta(\tau)}{q(B \mid X)\, q(\tau \mid X)\, q(Z \mid X)} \right]. \tag{66}$$

However, the importance-weighted ELBO (IWELBO) offers a tighter lower bound for evaluation. Using SLANTIS and the JC sampler, the IWELBO is computed as

$$\text{LL} = \mathbb{E}_{B', \tau' \sim s_\phi(B, \tau)}\left[ \log \frac{1}{L} \sum_{l=1}^{L} \frac{p_\theta(X, B^{(l)}, \tau^{(l)})}{s_\phi(B^{(l)}, \tau^{(l)})} \right], \tag{67}$$

where the sampling distribution may be factorized as

$$s_\phi(B, \tau) = s_\phi(B \mid \tau)\, s_\phi(\tau), \quad s_\phi(B \mid \tau) = \prod_{e \in E(\tau)} s_\phi(b(e)). \tag{68}$$

Note that the auxiliary variable $Z$ is marginalized out, as they do not compute LL using the variational distributions for $Z$.

### I.2.4 PHYLOGFN (ZHOU ET AL., 2024A)

PhyloGFN uses a GFlowNet-based sampler to approximate the posterior over phylogenetic trees and branch lengths. To evaluate how well the sampler approximates the true posterior, the marginal log-likelihood (MLL) is estimated using an importance-weighted variational lower bound:

$$\log P(Y) \geq \mathbb{E}_{\tau^{1:K} \sim P_F}\left[ \log \frac{P(z)\, \frac{1}{K} \sum_{i=1}^{K} P_B(\tau^i \mid z^i, b^i)\, R(z^i, b^i)}{P_F(\tau^i)} \right], \tag{69}$$

where $P_F$ is the GFlowNet policy over trajectories $\tau$, $P_B(\tau \mid z, b)$ is the branch length distribution conditioned on latent variables $z, b$, and $R(z, b)$ is a reweighting function.

### I.2.5 PHYLOGEN (DUAN ET AL., 2024)

PhyloGEN aims to maximize the expected marginal likelihood of the observed species sequences $Y$:

$$\max \log p(Y \mid (\tau(z), B_\tau)), \tag{70}$$

where $\tau(z)$ denotes the sampled tree topology and $B_\tau$ the corresponding branch lengths.

This is approximated using a variational distribution:

$$q(\tau(z), B_\tau \mid Y) = q(B_\tau \mid \tau(z))\, q(\tau(z)). \tag{71}$$

The joint probability is

$$p(Y, \tau(z), B_\tau) = p(Y \mid \tau(z), B_\tau)\, p(B_\tau \mid \tau(z))\, p(\tau(z)), \tag{72}$$

assuming conditional independence between the tree topology and branch lengths.

The Evidence Lower Bound (ELBO) is formulated as

$$\mathcal{L}(Q) = \mathbb{E}_q[\log P(Y, \tau(z), B_\tau)] - \mathbb{E}_q[\log q(\tau(z), B_\tau)]. \tag{73}$$

To improve training stability, a regularization term $R(z \mid \tau(z^*))$ is introduced:

$$\mathcal{L}(Q, R) = \mathbb{E}_{q(z)}\Big[\mathbb{E}_{q(B_\tau \mid \tau(z))}\big[\log P(Y, B_\tau \mid \tau(z))\, p(\tau(z))\, R(z \mid \tau(z^*))\big]\Big] - \mathbb{E}_q[\log q(\tau(z), B_\tau)]. \tag{74}$$

For reduced variance and better performance, a multi-sample approach is adopted:

$$\mathcal{L}_{\text{multi-sample}}(Q, R) = \frac{1}{K} \sum_{k=1}^{K} \log \big[ p(Y, B_{\tau_k} \mid \tau(z_k))\, p(\tau(z_k))\, R(z_k \mid \tau(z_k^*)) \big], \tag{75}$$

where $K$ is the number of Monte Carlo samples, and $z_k^*$, $B_{\tau_k}$ are sampled from $q(z^*)$ and $q(B_\tau \mid \tau(z))$, respectively.

## J ADDITIONAL RESULTS

### J.1 EXPERIMENTAL PROCEDURE - TREE TOPOLOGICAL DIVESITY ANALYSIS

For MrBayes, we first collect the top 2000 distinct trees across 10 independent MCMC runs for each dataset, similar to the procedure used during training. From these, we sample 1000 trees according to the posterior probabilities of each unique topology. This procedure ensures that the sampled set reflects the posterior distribution estimated by MrBayes, although it does not exactly replicate the software's internal sampling probabilities. For all other methods, we sample 1000 trees from each model. All trees are then sorted, unrooted, and metrics are computed on this standardized set.

### J.2 EXPERIMENTAL PROCEDURE - POSTERIOR COMPARISON

For MrBayes, we first collect all distinct trees topologies across 10 independent MCMC runs for each dataset. These runs retain all unique trees whose cumulative posterior sums to $1 - \epsilon$, with $\epsilon = 10^{-6}$. Branch lengths are added using our edge-length model. Let $\{\tau_i^{\text{MB}}, \mathbf{b}_i^{\text{MB}}\}_{i=1}^N$ and $\{\tau_i^{\text{PTD}}, \mathbf{b}_i^{\text{PTD}}\}_{i=1}^N$ denote the sampled MrBayes and PhyloTextDiff trees, respectively. For each model $M$, the mean unnormalized log-posterior is computed as

$$\overline{\log p}_M = \frac{1}{N} \sum_{i=1}^{N} \log \Big[ p(\mathbf{Y} \mid \tau_i^M, \mathbf{b}_i^M)\, p(\tau_i^M, \mathbf{b}_i^M) \Big]. \tag{76}$$

PhyloTextDiff trees are sampled for 21 hours, corresponding to the training time of MrBayes, and the unnormalized log-posterior distributions are visualized for comparison. This set up corresponds to experiment 1 in Table 6.

### J.3 TREE TOPOLOGICAL COMPARISON WITH MRBAYES

We performed an additional experiment to compare the tree space explored by MrBayes and Phylo-TextDiff using equal numbers of sampled trees. Table 6 summarizes the results for two experiments. For each model, we report the number of sampled trees, the number of unique trees, the number of trees shared between the two methods, and the mean posterior probability. We observe that Phylo-TextDiff consistently discovers more unique trees as the number of sampled trees increases. MrBayes' results are not directly comparable because we only have access to the output file with posterior values, which does not include very low probability trees. This means that the number of unique trees is an approximation, although only low probability trees are excluded from the calculations. Across both experiments, PhyloTextDiff achieves slightly higher mean posterior values on average.

Figure 8 shows the normalized histograms of log-posterior values for experiment 2. The distributions indicate that PhyloTextDiff samples a broader range of tree topologies and emphasizes higher-posterior regions compared to MrBayes.

| Experiment | Model | Sampled Trees | Unique Trees | Common Trees | Mean Posterior |
|---|---|---|---|---|---|
| 1 | MrBayes | 2921 | 2921 | 2363 | -7147.79 |
|  | PhyloTextDiff | 42109 | 11656 | – | -7146.64 |
| 2 | MrBayes | 2521 | 2521 | 915 | -7147.31 |
|  | PhyloTextDiff | 2521 | 1709 | – | -7146.91 |

Table 6: Comparison of MrBayes and PhyloTextDiff across Experiments 1 and 2.

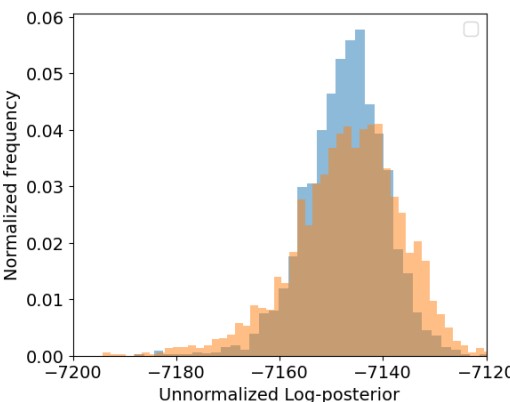

Figure 8: Normalized histograms of unnormalized log-posterior values for MrBayes (blue) and PhyloTextDiff (orange) on dataset DS1 for experiment 2. PhyloTextDiff (equal sampling) shows a less peaked distribution, indicating greater sampling diversity and better coverage of high-probability trees.

### J.4 PHYLOGENETIC CLUSTER ANALYSIS AND CHARACTERISTIC SPLITS

We analyzed phylogenetic trees generated by two methods: MrBayes and our PhyloTextDiff model after a sampling time of 21 hours. Clusters of similar trees were identified across the combined dataset. Let $\mathcal{C}_k$ denote the set of trees belonging to cluster $k$.

For each cluster, we first identified *splits*, i.e., bipartitions of leaf nodes induced by internal nodes, ignoring trivial splits (single leaves or the root). Characteristic splits were defined as those that occur frequently within a cluster but are rare outside the cluster. Formally, a split $s$ is characteristic of cluster $k$ if its *frequency within the cluster* $k$ $f_{\text{in},k}(s) \geq 0.5$ and its *frequency outside the cluster* $f_{\text{out},k}(s) \leq 0.2$. This ensures that characteristic splits capture distinctive phylogenetic patterns specific to a cluster.

To visualize the 8 clusters discovered by PhyloTextDiff and the cluster containing only MrBayes trees, we selected a representative tree from each cluster and highlighted its characteristic splits. They are depicted in Figures 9, 10, 11, 12 and 13. Each split was assigned a unique color, and both vertical and horizontal branches corresponding to the split were colored for emphasis. Node labels were retained in black to maintain readability. Table 7 summarizes the properties of the identified phylogenetic clusters. For each cluster, we report the cluster ID, the composition of trees generated by MrBayes and PhyloTextDiff, and the mean unnormalized posterior $\overline{\log p}$ across the trees in the cluster (see section J.2 for details on the computation of the unnormalized posterior). In addition, we list the characteristic splits identified within each cluster. Each split is assigned a color corresponding to the branches highlighted in the visualizations, along with its frequency within the cluster (*Inside*) and its frequency outside the cluster (*Outside*). Clusters often contain multiple characteristic splits, reflecting distinct, recurrent evolutionary groupings. High *Inside* frequencies coupled with low *Outside* frequencies indicate splits that are strongly representative of a particular cluster, making them useful markers for interpreting cluster-specific phylogenetic structure. This analysis demonstrates that PhyloTextDiff was able to discover novel high-probability phylogenetic trees that were not identified by MrBayes, highlighting its ability to efficiently explore alternative high-probability topologies.

Table 7: Comparison of clusters. For each cluster, the table shows the number of trees contributed by MrBayes and PhyloTextDiff, the mean unnormalized log-posterior, and the characteristic splits with their frequencies inside and outside the cluster.

| Fig | Cluster | $\{\tau_i^{\mathrm{MB}}\}/\{\tau_i^{\mathrm{PTD}}\}$ | $\overline{\log p}$ | Split | Inside | Outside |
|---|---|---|---|---|---|---|
| 9a | 1 | 0/27 | -7165.49 | Red | 0.93 | 0.12 |
| | | | | Blue | 0.74 | 0.03 |
| | | | | Green | 0.74 | 0.07 |
| | | | | Orange | 0.63 | 0.01 |
| | | | | Purple | 0.50 | 0.00 |
| | | | | Brown | 0.59 | 0.00 |
| 9b | 2 | 0/11 | -7169.42 | Red | 1.00 | 0.01 |
| | | | | Blue | 0.91 | 0.18 |
| | | | | Green | 0.82 | 0.15 |
| | | | | Orange | 0.55 | 0.18 |
| 10a | 3 | 0/34 | -7162.49 | Red | 0.59 | 0.07 |
| | | | | Blue | 0.50 | 0.13 |
| 10b | 4 | 0/17 | -7175.60 | Red | 0.88 | 0.18 |
| | | | | Blue | 0.59 | 0.00 |
| 11a | 5 | 0/11 | -7164.18 | Red | 1.00 | 0.17 |
| | | | | Blue | 0.64 | 0.05 |
| 11b | 6 | 0/8 | -7165.37 | Red | 0.88 | 0.05 |
| | | | | Blue | 0.62 | 0.05 |
| 12a | 7 | 0/25 | -7156.85 | Red | 1.00 | 0.18 |
| | | | | Blue | 0.96 | 0.17 |
| | | | | Green | 0.84 | 0.02 |
| 12b | 8 | 0/11 | -7157.77 | Red | 0.91 | 0.18 |
| | | | | Blue | 0.91 | 0.17 |
| 13 | 9 | 5/0 | -7178.03 | Red | 1.00 | 0.00 |
| | | | | Blue | 0.60 | 0.10 |

## J.5 RUNNING TIME

We evaluate the computational efficiency of our discrete diffusion model by reporting both training and sampling times (for 100 trees per dataset) for DS1–DS8 in Table 8. Our results demonstrate that the diffusion model can improve posterior approximation of tree topologies across all eight datasets in roughly 6 hours for the small model and 13 hours for the medium model.

We also report the running times (in minutes) for the edge-length model, corresponding to the training required to obtain our MLL/ELBO results (Table 9).

For comparison, we include the running times of PhyloGFN, the previous reproducible state-of-the-art method, in Table 10. These numbers are taken from the original paper. PhyloGFN was trained on virtual machines with 10 CPU cores and 10 GB of RAM for all datasets, and uses a single V100 GPU for DS1–DS6 and a single A100 GPU for DS7–DS8. Despite the differences in hardware, our discrete diffusion model consistently trains faster than PhyloGFN while achieving superior posterior approximation.

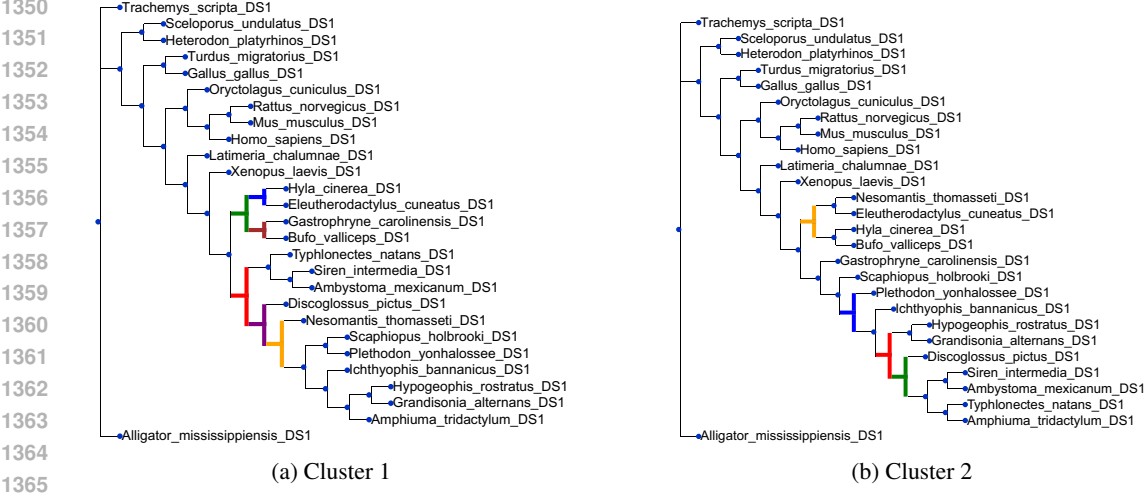

(a) Cluster 1

(b) Cluster 2

Figure 9: Representative trees from clusters 1 and 2

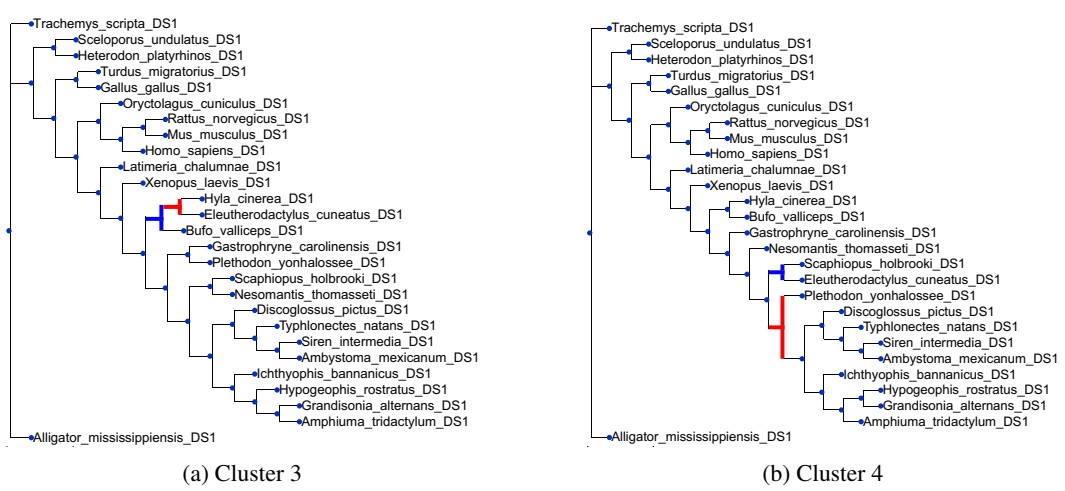

(a) Cluster 3

(b) Cluster 4

Figure 10: Representative trees from clusters 3 and 4

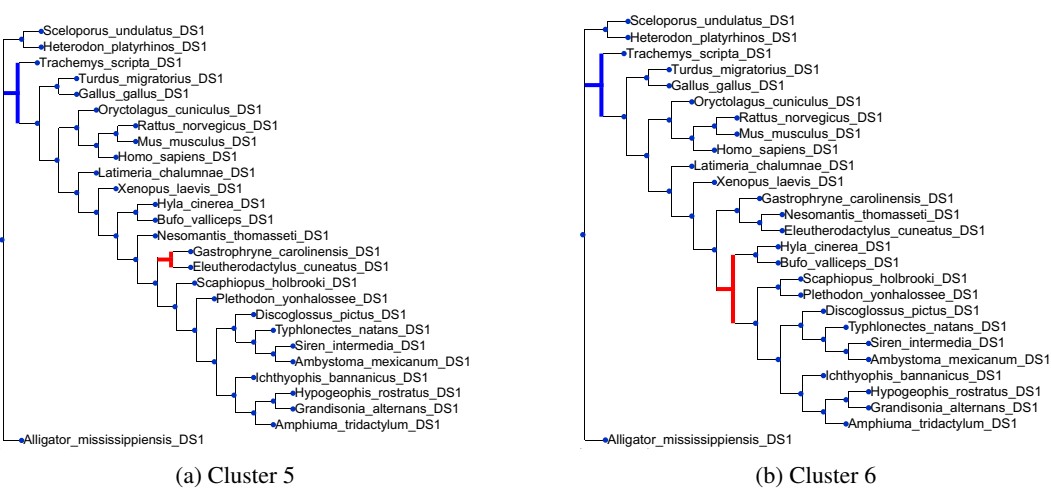

(a) Cluster 5

(b) Cluster 6

Figure 11: Representative trees from clusters 5 and 6

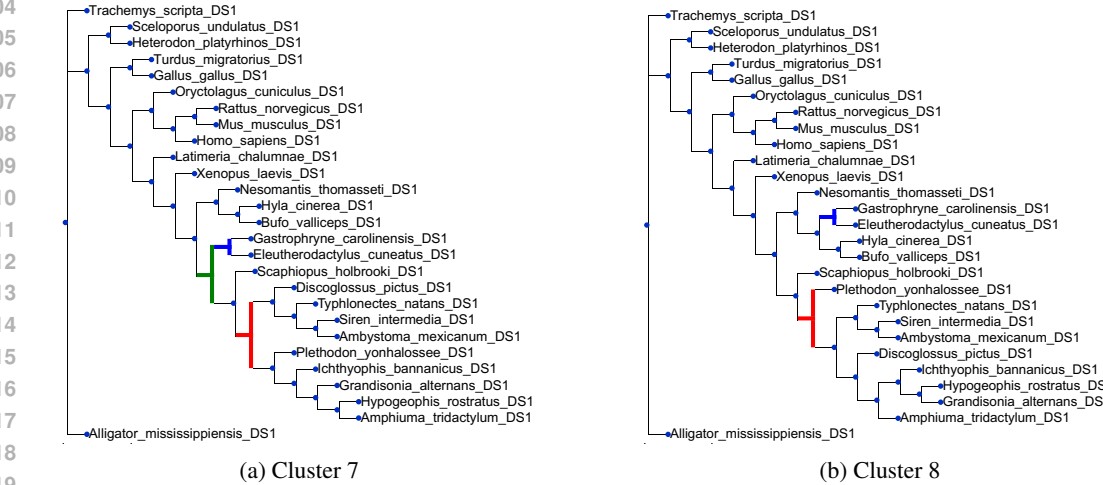

(a) Cluster 7              (b) Cluster 8

Figure 12: Representative trees from clusters 7 and 8

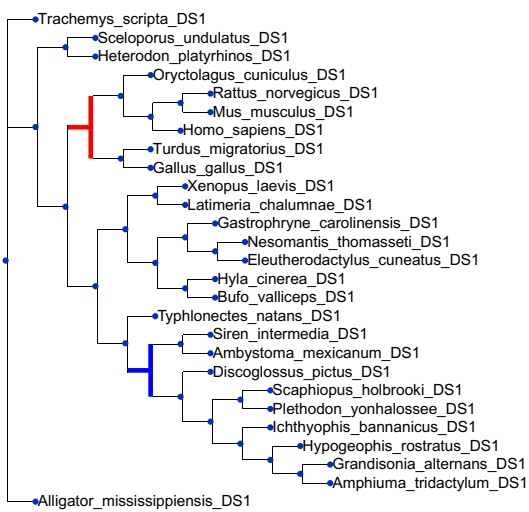

Figure 13: Representative tree from cluster 9

Table 9: Running time (minutes) and number of steps for the edge-length model for each of the eight datasets DS1-8

| Methods | Criteria | DS1 | DS2 | DS3 | DS4 | DS5 | DS6 | DS7 | DS8 |
|---|---|---|---|---|---|---|---|---|---|
| MrBayes + PhyloTextDiff | Time | 560.37 | 582.32 | 587.18 | 551.78 | 355.20 | 355.20 | 352.15 | 352.15 |
| MrBayes + PhyloTextDiff | Step | 1200 | 1400 | 1400 | 1000 | 800 | 800 | 800 | 800 |
| IQ-TREE + PhyloTextDiff | Time | 1430.65 | 1365.98 | 1377.28 | 1361.68 | 993.67 | 1187.62 | 1285.62 | 1392.97 |
| IQ-TREE + PhyloTextDiff | Step | 1000 | 1000 | 1200 | 800 | 400 | 600 | 1000 | 600 |
| VBPI-GNN + PhyloTextDiff | Time | 1251.55 | 1384.93 | 1326.92 | 1364.52 | 1394.90 | 1206.32 | 1328.12 | 1342.43 |
| VBPI-GNN + PhyloTextDiff | Step | 1000 | 1200 | 1200 | 1000 | 1000 | 800 | 1200 | 1000 |

Table 10: Running times (in minutes) of PhyloGFN for each of the eight datasets DS1–DS8.

| Method | DS1 | DS2 | DS3 | DS4 | DS5 | DS6 | DS7 | DS8 |
|---|---|---|---|---|---|---|---|---|
| PhyloGFN Full | 3760 | 4156 | 4820 | 6234 | 7670 | 8110 | 10443 | 11425 |
| PhyloGFN Short | 1240 | 1680 | 2140 | 2670 | 3100 | 3190 | 3620 | 3700 |

Table 8: Running time and sampling time of the diffusion model for the eight datasets DS1-8

| Method | Training Time | Sampling Time | Model size |
|---|---|---|---|
| MrBayes + PhyloTextDiff | 80k steps - 366.62 min | 10.73 min | Small |
| VBPI-GNN + PhyloTextDiff | 90k steps - 783.71 min | 33.27 min | Medium |
| IQ-Tree + PhyloTextDiff | 90k steps - 763.78 min | 33.92 min | Medium |

# K SENSITIVITY STUDY

We perform a sensitivity study on the number of denoising steps for each method, varying the steps among 512, 1024, 2048, and 4096. Table 11 reports the results, where the numbers correspond to a single-point estimate of the ELBO (using one particle, one run, and one repetition).

Across all datasets and all methods, the optimal number of denoising steps is 2048 steps, which is why we use this setting for evaluating our model. For training purposes, we choose 1024 steps, as it provides a good trade-off between performance and computation time.

Table 11: Sensitivity to the Number of Denoising Steps for PhyloTextDiff (ELBO). Best values are highlighted in **green**.

| Method | Denoising Steps | Mean Sampling Time | DS1 | DS2 | DS3 | DS4 | DS5 | DS6 | DS7 | DS8 |
|---|---|---|---|---|---|---|---|---|---|---|
| **MrBayes + PhyloTextDiff** | 512 | 14.94s | -7017.75 | -26273.10 | -33659.34 | -13281.47 | -8134.10 | -6639.81 | -37300.53 | -8556.10 |
| | 1024 | 28.61s | -6846.00 | -26110.11 | -33476.09 | -13056.23 | -7946.38 | -6458.04 | -37182.28 | -8391.58 |
| | 2048 | 45.55s | **-6347.73** | **-25619.47** | **-32971.85** | **-12582.51** | **-7437.71** | **-5956.42** | **-36674.04** | **-7899.23** |
| | 4096 | 113.79s | -6750.85 | -26032.95 | -33350.77 | -12983.59 | -7823.61 | -6349.53 | -37062.56 | -8278.81 |
| **IQ-Tree + PhyloTextDiff** | 512 | 32.49s | -7032.81 | -26290.19 | -33643.86 | -13254.67 | -8132.23 | -6860.49 | -37387.28 | -8563.92 |
| | 1024 | 93.51s | -6907.07 | -26092.58 | -33504.36 | -13074.31 | -7958.36 | -6476.03 | -37056.63 | -8380.22 |
| | 2048 | 98.94s | **-6396.94** | **-25673.42** | **-33146.95** | **-12594.38** | **-7476.04** | **-6043.45** | **-36604.53** | **-7906.54** |
| | 4096 | 242.20s | -6792.58 | -25997.40 | -33403.47 | -12981.94 | -7889.54 | -6580.53 | -37083.41 | -8323.76 |
| **VBPI-GNN + PhyloTextDiff** | 512 | 36.61s | -7026.37 | -26307.87 | -33645.66 | -13339.33 | -8123.20 | -6639.01 | -37259.26 | -8556.45 |
| | 1024 | 55.89s | -6822.00 | -26076.29 | -33448.41 | -13045.76 | -7934.72 | -6459.91 | -37056.63 | -8370.03 |
| | 2048 | 115.62s | **-6360.35** | **-25565.12** | **-32948.10** | **-12556.91** | **-7431.71** | **-5968.39** | **-36579.89** | **-7917.64** |
| | 4096 | 231.30s | -6732.86 | -26035.87 | -33403.02 | -12954.58 | -7867.63 | -6381.86 | -36998.67 | -8310.58 |

# L LIMITATIONS

While PhyloTextDiff achieves strong performance in modeling tree topology posteriors efficiently, several limitations remain. First, the model focuses exclusively on tree topologies and does not jointly model branch lengths in a fully generative way; instead, branch lengths are incorporated post hoc using a separate edge-length model. A natural extension would be to combine discrete diffusion for tree structures with continuous diffusion for branch lengths, or to design conditional architectures in which branch lengths are generated jointly with the evolving topology in an end-to-end framework (e.g., by predicting the parameters $\mu$ and $\sigma$ of the diagonal lognormal distribution). Second, the predefined tokenizer may introduce inductive bias, since taxa are assigned integer labels that could influence how the model learns relationships. This issue could be mitigated by adopting a one-hot encoding scheme or alternative representations that avoid imposing arbitrary orderings. Third, because of this tokenization design, the model cannot easily generalize to unseen taxa or datasets without retraining. A first potential solution would be to adopt a global vocabulary where each taxon is assigned a consistent token across datasets, rather than redefining tokens per dataset. Finally, the current model is tailored to DNA sequences and does not handle protein data or morphological traits. Extending the approach to broader data types would require integrating specialized encoders to replace the DNABERT module. Addressing these limitations is a key step toward building fully end-to-end generative models of phylogenetic trees.

## M    REPRODUCIBILITY

In this section, we describe the sampling procedures for different methods.

### M.1    MRBAYES

We obtained posterior samples of tree topologies by running 10 independent single-chain analyses in MrBayes (Ronquist et al., 2012), each for one billion iterations. Samples were collected every 1000 iterations, and the first 25% of each chain was discarded as burn-in.

The analyses were performed with the following command:

```
conda install bioconda::mrbayes

execute INPUT_FOLDER/ds1.nex
lset nst=1
prset statefreqpr=fixed(equal)
prset brlenspr=unconstrained:exp(10.0)
ss
ngen=1000000000
nruns=10
nchains=4
printfreq=1000
samplefreq=1000
savebrlens=yes
filename=OUTPUT_FOLDER
```

This setup specifies a Jukes-Cantor model (nst=1), a uniform prior over tree topologies and an i.i.d. exponential prior on branch lengths with rate 10.

### M.2    VBPI-GNN

We trained VBPI-GNN (Zhang, 2023) for datasets DS1–DS8 using the script provided on the paper's GitHub repository. For example, for DS1, the following command was executed:

```
python main.py --dataset DS1 --brlen_model gnn --gnn_type edge \
                --hL 2 --hdim 100 --maxIter 400000 --empFreq --psp
```

This setup specifies:

- a branch length model based on a GNN (`--brlen_model gnn`) with edge-based message passing (`--gnn_type edge`),
- 2 hidden layers (`--hL 2`) and 100 hidden units (`--hdim 100`),
- a maximum of 400,000 iterations (`--maxIter 400000`),
- empirical frequencies (`--empFreq`), and
- structured amortization using primary subsplit pairs (`--psp`).

### M.3    IQ-TREE

Phylogenetic inference was performed using IQ-TREE 2 (Minh et al., 2020). For dataset DS1, we ran:

```
iqtree2 -s DS1 -bb 10000 -wbt -m JC69 -redo
```

This command specifies the Jukes-Cantor substitution model (`-m JC69`) and performs 10,000 ultrafast bootstrap replicates (`-bb 10000`) to assess branch support. The `-wbt` flag writes branch lengths and support values to the tree files, and `-redo` ensures that any previous analyses are overwritten.

