# OpenReview forum: "PhyloTextDiff: Text-Based Discrete Diffusion for Generative Phylogenetic Inference"
_ICLR.cc/2026/Conference — Submitted to ICLR 2026_

### Official Review · Reviewer_s6Qi · 2025-10-31

**Soundness:** 1
**Presentation:** 2
**Contribution:** 4
**Rating:** 2
**Confidence:** 3

**Summary:**

The submission proposes a new diffusion-based method for sampling phylogenies proportional to the posterior. The diffusion model works on a data distribution over Newick representations of phylogenies drawn from MrBayes. The results reported are 100s of nats better in terms of marginal log-likelihood compared to the competitors.

**Strengths:**

The paper is very interesting and the idea of learning diffusion models to sample from phylogenetic posteriors is new, and a great fit to the ICLR commuties interests. There are multiple analyses of the results based on the standard phylogenetic benchmarking datasets (DS1-DS8) and the paper includes many baseline results.

An extra note on originality: the choice of doing generative modeling based on Newick representations of phylogenetic samples was positively surprising to me.

I have some worries about the exposition of the paper and the MLL scores reported, but I sincerely hope that we can sort it out during a discussion period as the method is an interesting contribution to the Bayesian/probabilistic phylogenetic inference community.

**Weaknesses:**

My score is currently quite low due to the many concerns/questions I have below. Hopefully we can sort it out, but in general the quality of writing (correctness, inclusion of key definitions, motivations, typos) is currently not sufficient for ICLR standards (I have a lot of suggested improvements below), and I need more evidence that the MLL is correctly computed due to the extreme scores. If my concerns are addressed, I am happy to consider raising my score.

## Motivation
I am missing the motivation of two key components of the submission.

First, why learn a diffusion model between noise and Newick string? That is, why not do it on some other representation of trees (e.g. via the subsplit representation in SBNs)? Are text-based diffusion models necessarily better than those that operate on other types of discrete data? This modeling choice deserves at least a motivation.

Why would we want a tree sampler that has been trained on DNA data and trees from different datasets? I see that there are generalizable patterns to be used in the construction of trees based on DNA that might be useful, but can you give some technical insights how this works? If you have two datasets, one with 10 vertebrate species and one with 1,000 viruses, is it still sensible to learn an amortization?

On the same topic, how would PhyloTextDiff perform if it was only trained on one specific dataset? How many datasets do we need for it to learn something meaningful, or to learn at all?

## Correctness

### The marginal log-likelihood
The **notation** in Eq. 8 is incorrect in the sense that the subscripts under the sum inside the log suggests that samples are drawn inside the log, while the expectation sits outside the log. Please clarify that the $K$ importance samples are drawn i.i.d.

There are some points that I want to sort out regarding the MLL computation:

How is the MLL is computed? $K$ trajectories $x_{0:T}$ and branch lengths are drawn from the samplers and then logsumexp() of the $K$ samples -$\log K$?

When sampling $x_{0:T}$, how is this done? I see the phrase ”base phylogenetic sampler” which is not precised in the main text. How is $x_T$ sampled in the generation?  The reason why I am zooming in on this is because I find the results in Table 2 to be unrealistically strong — I was under the impression that the MLL scores had saturated and could not be improved (only marginally, maybe), and I have seen that other reviewers on OpenReview have had the same opinion. I.e. that potentially the true marginal log-likelihoods had been achieved via the fact that when $K \rightarrow \infty$ then the IW-ELBO goes to the true marginal log-likelihood.

Now, the improvements in scores here are so extreme that I would like some certification that the implementation is correct. There is no code available at the moment, so that option is not available. The derivation of the IW-ELBO in the appendix looks correct, but 1) I am not an expert on diffusion models, so it would be good with inputs from other reviewers, and 2) some aspects (like how $x_{0:T}$ is actually sampled) is not described in the paper, as far as I can see.

Could you please explain this, for example: from the long-run MCMC samples on DS1 provided in both Molén et al. (2024; see Fig. 8 in https://openreview.net/pdf?id=TBLMrHaFFH) and in Whidden and Matsen IV (2015), there is not much diversity to speak of in terms of sampling diversity — MrBayes samples what looks like <50 unique tree-topologies. Given then the results in Fig. 4 (where the distribution of log-probabilities are largely similar to those from MrBayes), what are the reasons for the extreme improvements in MLL?


### MCMC speed statement

Line 41: ”In practice, however, MCMC suffers from slow convergence[…]” This is a statement in ML-based phylogenetic inference that is being thrown around without proper justification: I have not seen any compelling evidence that, e.g., VI-based, GFN-based or any other approach is as fast as MrBayes while performing on par according to relevant metrics. Yes, MrBayes is a software that has been optimized for a long time, but still, promises of improving runtimes with ML techniques are not convincingly supported, as far as I am aware. Thus, I think this statement, and ”VI methods provide faster alternatives” should be rephrased as I do not find them correct.

### Sequential/incremental tree building algorithms

There are multiple incorrect classifications of algorithms as  ”building” trees sequentially or not. Please either crystallize what is meant by ”incremental” such that the classifications are correct, or completely refrain from this approach to segmenting methods.

For instance, after the ”build trees incrementally” statement in line 46, please include SMC-based algorithms, too. In fact, are not all related works building trees incrementally? SBNs build from root to leaves, while PhyloGFN build from leaves to root. PhyloVAE also reconstructs tree ”incrementally” through a sequence of merges. In summary, I think the phrasing should be revised. I am not sure what methodologically unites (Xie and Zhang, 2023; Zhou et al., 2024a; Koptagel et al., 2023; Mimori and Hamada, 2023) so I cannot provide a suggestion, unfortunately.

Again, in line 90: ”Recent incremental approaches such as ArTree (Xie and Zhang, 2023) and PhyloGFN (Zhou et al., 2024a) build trees sequentially”, all methods build trees sequentially? SBNs, too, through a sequence of probability distributions over possible splits.

### Misc
”efficiently sample from a distribution that matches the posterior **as closely as possible**” this is arguably overly informal. If there is no way of bounding the expected quality of the sampling approximation, I recommend saying ”efficiently sample from a distribution that matches the posterior”.

In line 153-154: what is the distinction between (i), (ii) and (iii)? Goal (i) implies both (ii) and (iii), unless there is some trade-off between the goals that you would rather explore multiple modes than ”efficiently sample from a distribution that matches the posterior **as closely as possible**”. Is there a trade-off?

Here it reads as if the lower bound on the MLL is used to measure diversity, which is not an appropriate measure for this: ”Ideally, the approach should achieve high diversity, i.e., sampling a large number of unique trees. In order to measure the extent to which we achieve these goals, we evaluate lower bounds on the marginal log-likelihood (MLL) as well as several diversity measures, including Simpson’s diversity metric.” I would either remove the first sentence, or put it after the second one to avoid this confusion.

Furthermore, Simpson’s diversity metric is not defined and no pointer to a definition is provided.

$q_F$ and $q_{R,j}$ in Eq. 8 are not defined in the main text.

## Related work
There is plenty of interest in diversity and exploration of the phylogenetic tree space: this is very much the topic in Molén et al. (2024) where mixtures are used to efficiently explore the space. I would have expected to see this reference at least mentioned, but potentially also benchmarked against wrt the Simpson’s diversity metric.

## Writing
The technical explanations/expositions can be improved quite a bit. For instance:

The explanation of Newick formats is not easily parsed in line 109-111.

The exposition in Sec. 3.2 does appears quite rushed. There are multiple typos (see below), and $q_\text{base}$ and $q_\text{data}$ are not used in Eq. 1. Also, how the conclusion after ”Therefore” in line 137 was reached is not possible to understand, and the result seems important enough to not be passed to the appendix? Overall, this section needs more information and the writing needs to be improved.

Very importantly, in Eq. 8, $q_F$ and $q_{R,j}$ are not defined in the text (and no pointer to their definition is provided).

### Typos:
* The Zhang and Iv, 2019 reference in line 45 needs to updated.
* Line 87: Inconsistencies in capitalized letters: Variational inference and Generative Models
* Line 90: ArTree is misspelled -> ARTree
* Line 132-133: Missing spacing between, first, $o(\Delta t)$ and (Austin et al. 2021), and then ”(Austin et al., 2021).This”
* ”Generative Model” in line 159. Words in paragraph titles do not have capitalized first letters.
    * ”Baselines and Performance Metrics.”
    * Tree Topological Diversity Analysis
* Inconsistencies in referring to DMs: diffusion models or Diffusion Models.
* Line 202: ”DNABERT-S”, it is called DNABERT  as in Figure 2, no?
* Line 221: Capitalized first letters in paragraph title: ”Forward and Backward Process”
* Line 362: there are multiple typos in this paragraph, e.g. ”Artree”

**Questions:**

I have integrated plenty of questions in the previous cell.

---

### Official Review · Reviewer_EqYY · 2025-11-01

**Soundness:** 3
**Presentation:** 3
**Contribution:** 3
**Rating:** 4
**Confidence:** 4

**Summary:**

This paper introduces PhyloTextDiff, a discrete diffusion model that operates on Newick-formatted phylogenetic trees. The method trains on trees sampled from existing phylogenetic inference tools (MrBayes, VBPI-GNN, or IQ-TREE) and uses a transformer-based architecture with DNA embeddings to learn tree topology distributions. The authors report improved marginal log-likelihood (MLL) estimates and higher topological diversity compared to baselines across eight benchmark datasets.

**Strengths:**

Novel application of discrete diffusion to phylogenetics with solid technical execution. This paper represents the first successful application of discrete diffusion models to phylogenetic tree inference, adapting recent advances in text-based diffusion (Lou et al., 2024) to evolutionary tree reconstruction. The technical components—DNABERT embeddings for conditioning on genetic data, absorbing-state diffusion for Newick syntax constraints, and the importance-weighted lower bound derivation—demonstrate competent implementation. The ELBO comparisons in Table 2 show consistent improvements over strong baselines like VBPI-GNN across all datasets, suggesting genuine contributions beyond simple resampling.

**Weaknesses:**

* Unsubstantiated cross-dataset learning and missing ablations. The paper claims "cross-dataset learning", but the dataset-conditional loss and dataset-specific tokenization (Appendix C) suggest this is multi-task learning with shared parameters rather than true transfer learning. No evidence is provided that joint training outperforms independently trained models, nor are there transfer experiments (e.g., train on DS1-7, test on DS8). Architectural choices (RoPE, cross-attention, timestep embedders) lack justification—Appendix K only ablates denoising steps. The paper also omits comparison with PhyloVAE, the most relevant generative baseline mentioned in Related Work.

* Incomplete performance characterization. While diversity metrics improve (Table 3), it remains unclear whether this reflects better exploration of high-quality regions or dilution across low-probability trees. What fraction of PhyloTextDiff samples exceed MrBayes's highest posterior probability? Cluster 9 in Table 7 (5 MrBayes trees, 0 PhyloTextDiff trees) suggests potential sampling biases, but the paper does not verify whether this cluster was in the training data. The dependency on base samplers for training should be more prominently stated in the abstract.

* Computational cost and reproducibility concerns. The paper does not report training time, GPU requirements, memory usage, or the number of denoising steps used in final experiments. While Table 2 compares against MrBayes after 21 hours of sampling, the total wall-clock time for PhyloTextDiff (training + inference) is not disclosed, making it impossible to assess the method's practical efficiency. For reproducibility, the paper should report GPU-hours for training on each dataset and per-sample generation time.

**Questions:**

* What is PhyloTextDiff's performance when trained on randomly generated trees instead of MrBayes samples? This would test whether the model learns genuine phylogenetic structure or merely mimics the base sampler's distribution.

* What fraction of PhyloTextDiff samples have higher posterior probability than MrBayes's highest-probability tree? This would clarify whether increased diversity reflects better exploration or dilution across low-quality regions.

* Cluster 9 contains 5 MrBayes trees but zero PhyloTextDiff trees. Was this cluster included in the training data? If yes, why can't the model reproduce it? If no, does this suggest the model cannot extrapolate to unseen low-probability regions?

---

### Official Review · Reviewer_pBbB · 2025-11-02

**Soundness:** 3
**Presentation:** 2
**Contribution:** 2
**Rating:** 4
**Confidence:** 4

**Summary:**

This paper integrates the phylogenetic tree modeling with the discrete diffusion models, showing the potential of generating tree topologies non-autoregressively in a few steps.
The most important ingredients of the current methodology include the Newick sequential representation of the tree, the tree tokenizer, and the discrete diffusion over the tokenized sequence.
This method also shows the potential for joint learning for multiple data sets, while the current experimental setting remains unclear.
The experiments demonstrate the PhyloTextDiff method can improve diversities and provides reliable posterior estimation.

**Strengths:**

- This paper presents the first step of using the popular discrete diffusion models for phylogenetic tree modeling, and introduces DNABERT representation as the input, instead of relying on an order encoding.
- The experiments are elaborate, especially the posterior probability comparison to MrBayes, which shows the method works and gives more diverse and reliable estimation for the posterior.

**Weaknesses:**

- The cross-dataset training claim is not very convincing. Although the discrete diffusion can operate on multiple data sets, the GNNs for branch lengths are still dataset-specific (Line 197, line 205). Moreover, the branch length modeling still relies on an order encoding instead of the learnable DNA representation (e.g., from DNABERT).
- The tokenizer in Line 212 is not applicable for more general inference. This tokenizer directly gathers all the species names in the training data set and cannot adapt to the inference setting when there are species not in the training set. I understand this is not touched in the current experiment, but the impossibility of a more general inference will query of the necessity of joint training on multiple data sets.
- The experiments are performed on the unrooted trees. However, the Newick representations for unrooted trees are not unique and depend on the root position (i.e., the degree-3 node). How do the authors solve this issue?
- The "adaptation to a structured space" paragraph in line 266 remains ambiguous. The newick representation has a fixed number of commas and paired parentheses. How do you ensure the generating process satisfies these constraints?
- The questionable likelihood estimation results. In my opinion, the likelihood lower bound in Section 4.3 can be generally computed as $$L_K = \log (\frac{1}{K}\sum_{i=1}\frac{P(Y,x_0^i,b^i)}{Q_\theta(x_0^i,b^i|Y)})$$
where $P$ is the unnormalized posterior, and $Q$ is the model-based tree probability.
However,  on DS1, Figure 4 shows that the trees have a log posterior $P$ around -7100, and Table 2 reports $L_1$ is around -6400. This implies that $Q(x_0,b|Y)$ should be around $e^{-700}$, which is impossible for high-density trees. Given all other methods produce similar ELBO estimates, which deviate largely from PhyloTextDiff in Table 2, I think the authors should elaborate more detailedly how they compute the values in Table 2.
- In the abstract, the authors claim "text diffusion enabling fast and scalable generation that is minimally impacted by the number of taxa." However, they use a $T=1024$ sampling step, which is larger than the Newick sequence length. Given this, I think an important baseline should be the autoregressive generation of the Newick sequence, at least a comparison on the generation speed. Such an experiment would strengthen the claim in the abstract.
- As a continuation of the above point, it is worth investigating the trade-off between efficiency and accuracy by tuning $T$.

**Questions:**

- What is the learnable embedding $E$ in Line 292?

---

### Official Review · Reviewer_PLL7 · 2025-11-03

**Soundness:** 1
**Presentation:** 3
**Contribution:** 2
**Rating:** 2
**Confidence:** 4

**Summary:**

This contribution introduces PhyloTextDiff (PTD), a neural architecture for amortized inference of posterior distributions of phylogenetic trees. PTD relies on a discrete diffusion process to learn tree distributions from a training sample obtained from a sampling method such as MCMC. The diffusion is applied to a newick (character string) encoding of the tree, and is conditioned on a set of DNA sequences through a pre-trained DNABERT encoding. PTD is compared to existing variational inference methods on a standard benchmark.

**Strengths:**

The article is generally well-written and organized, and the problem of making Bayesian phylogenetic inference faster and more accurate is both difficult and important.

**Weaknesses:**

I am mostly concerned by the evaluation of the method. PTD is meant to learn a posterior distribution from a sample obtained from another method (eg MrBayes or VBPI-GNN). Its training does not optimize any other objective so it seems like the best it could do is match this distribution. It is for example surprising that it outperforms MrBayes in MLL (Table 1). The introduction claims that the diffusion process refines existing posteriors, but I don't see how it would do it.

I may be missing something but is seems like different MLL lower bounds were used for different methods, which may explain this result but which in my opinion makes it difficult to conclude anything from this comparison. I also do not understand how ELBO was computed for PTD since it is stated that it cannot be evaluated (Appendix D). Since no code was provided, it is difficult to check what was really done.

**Questions:**

A few other points were unclear to me:

- The conditioning on Y is done through pre-trained DNABERT
  embeddings. I don't understand the rationale for this since the
  training data are sampled under a JC model of DNA sequence evoluion,
  which is site-independent. Isn't it guaranteed that the training
  data has no co-evolution, and that any contextual signal picked up
  by the embeddings is noise?

- How is it ensured that the newick strings output by PTD are
  correctly formatted? I don't see how there is a guarantee that they
  produce strings corresponding to a tree. If this is not guaranteed,
  how does the method deal with incorrectly formed outputs?

- PTD is trained over samples from MrBayes, VBPI-GNN or
  IQTREE-bootstrap. I don't think bootstraps produce samples from the
  posterior distribution, so what is the motivation for this choice,
  and what it the intuition behind the fact that it outperforms
  MrBayes in MLL and both MrBayes and VBPI-GNN in ELBO (both MLL and
  ELBO are supposed to measure the fit to the posterior)?

- The newick-based approach is motivated by scaling to larger trees,
  but the string scales in nmax (largest number of leaves in training)
  and goes through transformer layers. Isn't the resulting complexity
  quadratic in nmax? If not, it would help to detail the memory and
  runtime complexity. It is likely that poor scalings would not show
  on the presented experiments because nmax=64.

---

### Author Response · Authors · 2025-11-12
**Code Availability for Review**

Dear reviewers,

Thank you for your valuable feedback. I will take some time to address all the comments carefully. Meanwhile, please find the code at the following link for your reference: https://anonymous.4open.science/r/PhyloTextDiff-C2B5/

---

### Meta-Review · Area_Chair_xyj3 · 2026-01-04

**Summary:**

The paper received an average rating of 3.0. Specifically, all reviewers were leaning towards rejection in the original reviews (2, 2, 4, 4). The authors did not provide a rebuttal to address the reviewers' valuable questions and concerns. Thus, the AC recommended a rejection.

**Reviewer Concerns:**

No rebuttal is provided.

**Reviewer Scores:**

No rebuttal is provided.

---

### Decision · Program_Chairs · 2026-01-26

Reject